

# A brief introduction to mixed effects modelling and multi-model inference in ecology

Xavier A. Harrison[1], Lynda Donaldson[2,3], Maria Eugenia Correa-Cano[2], Julian Evans[4,5], David N. Fisher[4,6], Cecily E.D. Goodwin[2], Beth S. Robinson[2,7], David J. Hodgson[4] and Richard Inger[2,4]

[1] Institute of Zoology, Zoological Society of London, London, UK
[2] Environment and Sustainability Institute, University of Exeter, Penryn, UK
[3] Wildfowl and Wetlands Trust, Slimbridge, Gloucestershire, UK
[4] Centre for Ecology and Conservation, University of Exeter, Penryn, UK
[5] Department of Biology, University of Ottawa, Ottawa, ON, Canada
[6] Department of Integrative Biology, University of Guelph, Guelph, ON, Canada
[7] WildTeam Conservation, Padstow, UK

Corresponding author
Xavier A. Harrison,
x.harrison@ucl.ac.uk

## ABSTRACT

The use of linear mixed effects models (LMMs) is increasingly common in the analysis of biological data. Whilst LMMs offer a flexible approach to modelling a broad range of data types, ecological data are often complex and require complex model structures, and the fitting and interpretation of such models is not always straightforward. The ability to achieve robust biological inference requires that practitioners know how and when to apply these tools. Here, we provide a general overview of current methods for the application of LMMs to biological data, and highlight the typical pitfalls that can be encountered in the statistical modelling process. We tackle several issues regarding methods of model selection, with particular reference to the use of information theory and multi-model inference in ecology. We offer practical solutions and direct the reader to key references that provide further technical detail for those seeking a deeper understanding. This overview should serve as a widely accessible code of best practice for applying LMMs to complex biological problems and model structures, and in doing so improve the robustness of conclusions drawn from studies investigating ecological and evolutionary questions.

## INTRODUCTION

In recent years, the suite of statistical tools available to biologists and the complexity of biological data analyses have grown in tandem (*Low-Décarie, Chivers & Granados, 2014*; *Zuur & Ieno, 2016*; *Kass et al., 2016*). The availability of novel and sophisticated statistical techniques means we are better equipped than ever to extract signal from noisy biological data, but it remains challenging to know how to apply these tools, and which statistical technique(s) might be best suited to answering specific questions

(*Kass et al., 2016*). Often, simple analyses will be sufficient (*Murtaugh, 2007*), but more complex data structures often require more complex methods such as linear mixed effects models (LMMs) (*Zuur et al., 2009*), generalized additive models (*Wood, Goude & Shaw, 2015*) or Bayesian inference (*Ellison, 2004*). Both accurate parameter estimates and robust biological inference require that ecologists be aware of the pitfalls and assumptions that accompany these techniques and adjust modelling decisions accordingly (*Bolker et al., 2009*).

Linear mixed effects models and generalized linear mixed effects models (GLMMs), have increased in popularity in the last decade (*Zuur et al., 2009*; *Bolker et al., 2009*). Both extend traditional linear models to include a combination of fixed and random effects as predictor variables. The introduction of random effects affords several non-exclusive benefits. First, biological datasets are often highly structured, containing clusters of non-independent observational units that are hierarchical in nature, and LMMs allow us to explicitly model the non-independence in such data. For example, we might measure several chicks from the same clutch, and several clutches from different females, or we might take repeated measurements of the same chick's growth rate over time. In both cases, we might expect that measurements within a statistical unit (here, an individual, or a female's clutch) might be more similar than measurements from different units. Explicit modelling of the random effects structure will aid correct inference about fixed effects, depending on which level of the system's hierarchy is being manipulated. In our example, if the fixed effect varies or is manipulated at the level of the clutch, then treating multiple chicks from a single clutch as independent would represent pseudoreplication, which can be controlled carefully by using random effects. Similarly, if fixed effects vary at the level of the chick, then non-independence among clutches or mothers could also be accounted for. Random effects typically represent some grouping variable (*Breslow & Clayton, 1993*) and allow the estimation of variance in the response variable within and among these groups. This reduces the probability of false positives (Type I error rates) and false negatives (Type II error rates, e.g. *Crawley, 2013*). In addition, inferring the magnitude of variation within and among statistical clusters or hierarchical levels can be highly informative in its own right. In our bird example, understanding whether there is more variation in a focal trait among females within a population, rather than among populations, might be a central goal of the study.

Linear mixed effects models are powerful yet complex tools. Software advances have made these tools accessible to the non-expert and have become relatively straightforward to fit in widely available statistical packages such as R (*R Core Team, 2016*). Here we focus on the implementation of LMMs in R, although the majority of the techniques covered here can also be implemented in alternative packages including SAS (SAS Institute, Cary, NC, USA) & SPSS (SPSS Inc., Chicago, IL, USA). It should be noted, however, that due to different computational methods employed by different packages there may be differences in the model outputs generated. These differences will generally be subtle and the overall inferences drawn from the model outputs should be the same.

Despite this ease of implementation, the correct use of LMMs in the biological sciences is challenging for several reasons: (i) they make additional assumptions about the data to those made in more standard statistical techniques such as general linear models (GLMs), and these assumptions are often violated (*Bolker et al., 2009*); (ii) interpreting model output correctly can be challenging, especially for the variance components of random effects (*Bolker et al., 2009*; *Zuur et al., 2009*); (iii) model selection for LMMs presents a unique challenge, irrespective of model selection philosophy, because of biases in the performance of some tests (e.g. Wald tests, Akaike's Information Criterion (AIC) comparisons) introduced by the presence of random effects (*Vaida & Blanchard, 2005*; *Dominicus et al., 2006*; *Bolker et al., 2009*). Collectively, these issues mean that the application of LMM techniques to biological problems can be risky and difficult for those that are unfamiliar with them. There have been several excellent papers in recent years on the use of GLMMs in biology (*Bolker et al., 2009*), the use of information theory (IT) and multi-model inference for studies involving LMMs (*Grueber et al., 2011*), best practice for data exploration (*Zuur et al., 2009*) and for conducting statistical analyses for complex datasets (*Zuur & Ieno, 2016*; *Kass et al., 2016*). At the interface of these excellent guides lies the theme of this paper: an updated guide for the uninitiated through the model fitting and model selection processes when using LMMs. A secondary but no less important aim of the paper is to bring together several key references on the topic of LMMs, and in doing so act as a portal into the primary literature that derives, describes and explains the complex modelling elements in more detail.

We provide a best practice guide covering the full analysis pipeline, from formulating hypotheses, specifying model structure and interpreting the resulting parameter estimates. The reader can digest the entire paper, or snack on each standalone section when required. First, we discuss the advantages and disadvantages of including both fixed and random effects in models. We then address issues of model specification, and choice of error structure and/or data transformation, a topic that has seen some debate in the literature (*O'Hara & Kotze, 2010*; *Ives, 2015*). We also address methods of model selection, and discuss the relative merits and potential pitfalls of using IT, AIC and multi-model inference in ecology and evolution. At all stages, we provide recommendations for the most sensible manner to proceed in different scenarios. As with all heuristics, there may be situations where these recommendations will not be optimal, perhaps because the required analysis or data structure is particularly complex. If the researcher has concerns about the appropriateness of a particular strategy for a given situation, we recommend that they consult with a statistician who has experience in this area.

## UNDERSTANDING FIXED AND RANDOM EFFECTS

A key decision of the modelling process is specifying model predictors as fixed or random effects. Unfortunately, the distinction between the two is not always obvious, and is not helped by the presence of multiple, often confusing definitions in the literature (see *Gelman & Hill, 2007*, p. 245). Absolute rules for how to classify something as a fixed or random effect generally are not useful because that decision can change depending on

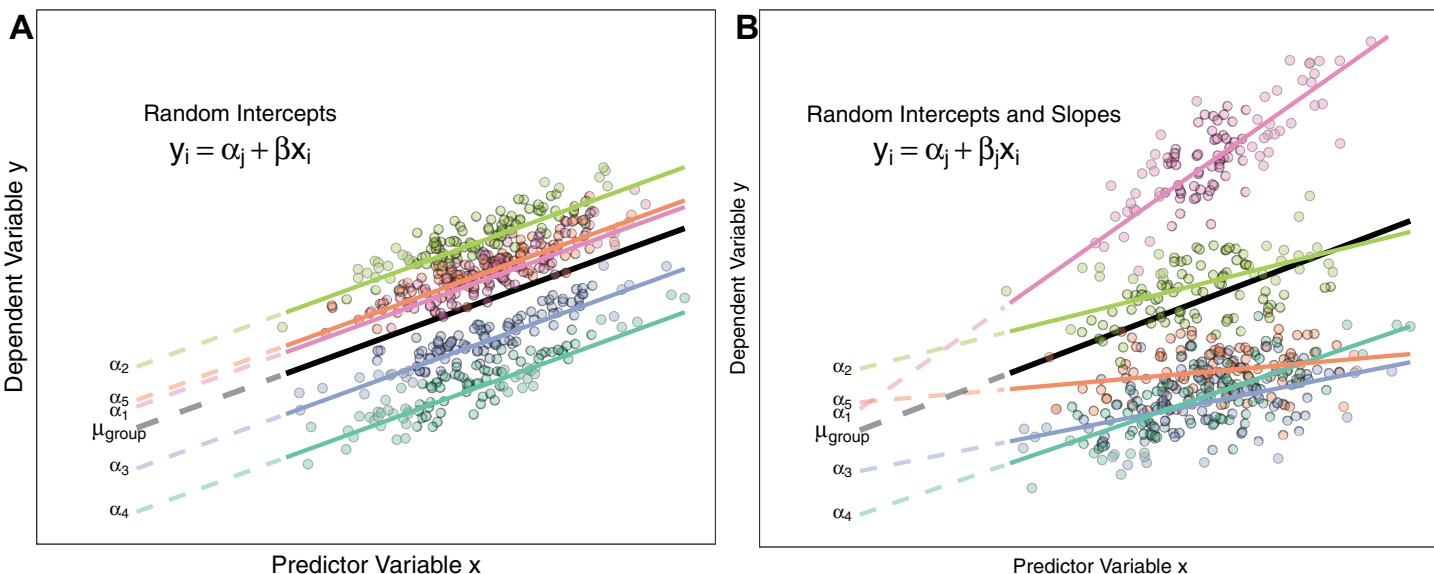

**Figure 1 Differences between Random Intercept vs Random Slope Models.** (A) A random-intercepts model where the outcome variable *y* is a function of predictor *x*, with a random intercept for group ID (coloured lines). Because all groups have been constrained to have a common slope, their regression lines are parallel. Solid lines are the regression lines fitted to the data. Dashed lines trace the regression lines back to the *y* intercept. Point colour corresponds to group ID of the data point. The black line represents the global mean value of the distribution of random effects. (B) A random intercepts and random slopes model, where both intercepts and slopes are permitted to vary by group. Random slope models give the model far more flexibility to fit the data, but require a lot more data to obtain accurate estimates of separate slopes for each group.

the goals of the analysis (*Gelman & Hill, 2007*). We can illustrate the difference between fitting something as a fixed (M1) or a random effect (M2) using a simple example of a researcher who takes measurements of mass from 100 animals from each of five different groups ($n = 500$) with a goal of understanding differences among groups in mean mass. We use notation equivalent to fitting the proposed models in the statistical software R (*R Core Team, 2016*), with the LMMs fitted using the R package *lme4* (*Bates et al., 2015b*):

```
M1 <- lm (mass ~ group)
M2 <- lmer(mass ~ 1 + (1|group)
```

Fitting 'group' as a fixed effect in model M1 assumes the five 'group' means are all independent of one another, and share a common residual variance. Conversely, fitting group as a random intercept model in model M2 assumes that the five measured group means are only a subset of the realised possibilities drawn from a 'global' set of population means that follow a Normal distribution with its own mean ($\mu_{group}$, Fig. 1A) and variance ($\sigma^2_{group}$). Therefore, LMMs model the variance hierarchically, estimating the processes that generate among-group variation in means, as well as variation within groups. Treating groups from a field survey as only a subset of the *possible* groups that could be sampled is quite intuitive, because there are likely many more groups (e.g. populations) of the study species in nature than the five the researcher measured. Conversely if one has designed an experiment to test the effect of three different

temperature regimes on growth rate of plants, specifying temperature treatment as a fixed effect appears sensible because the experimenter has deliberately set the variable at a given value of interest. That is, there are no unmeasured groups with respect to that particular experimental design.

Estimating group means from a common distribution with known (estimated) variance has some useful properties, which we discuss below, and elaborate on the difference between fixed and random effects by using examples of the different ways random effects are used in the literature.

## Controlling for non-independence among data points

This is one of the most common uses of a random effect. Complex biological data sets often contain nested and/or hierarchical structures such as repeat measurements from individuals within and across units of time. Random effects allow for the control of non-independence by constraining non-independent 'units' to have the same intercept and/or slope (*Zuur et al., 2009*; *Zuur & Ieno, 2016*). Fitting *only* a random intercept allows group means to vary, but assumes all groups have a common slope for a fitted covariate (fixed effect). Fitting random intercepts *and* slopes allows the slope of a predictor to vary based on a separate grouping variable. For example, one hypothesis might be that the probability of successful breeding for an animal is a function of its body mass. If we had measured animals from multiple sampling sites, we might wish to fit 'sampling site' as a random intercept, and estimate a common slope (change in breeding success) for body mass across all sampling sites by fitting it as a fixed effect:

```
M3 <- glmer(successful.breed ~ body.mass + (1|sample.site),
family=binomial)
```

Conversely, we might wish to test the hypothesis that the strength of the effect (slope) of body mass on breeding success varies depending on the sampling location i.e. the change in breeding success for a 1 unit change in body mass is not consistent across groups (Fig. 1B). Here, 'body mass' is specified as a random slope by adding it to the random effects structure. This model estimates a random intercept, random slope, and the correlation between the two and also the fixed effect of body mass:

```
M4 <- glmer(successful.breed ~ body.mass + (body.mass|sample.site),
family=binomial)
```

*Schielzeth & Forstmeier (2009)*; *Barr et al. (2013)* and *Aarts et al. (2015)* show that constraining groups to share a common slope can inflate Type I and Type II errors. Consequently, *Grueber et al. (2011)* recommend always fitting both random slopes and intercepts where possible. Whether this is feasible or not will depend on the data structure (see 'Costs to Fitting Random Effects' section below). Figure 1 describes the differences between random intercept models and those also containing random slopes.

*Further reading: Zuur & Ieno (2016) shows examples of the difficulties in identifying the dependency structure of data and how to use flow charts/graphics to help decide model structure. Kéry (2010, Ch 12) has an excellent demonstration of how to fit random slopes, and*

*how model assumptions change depending on specification of a correlation between random slopes and intercepts or not. Schielzeth & Forstmeier (2009) and van de Pol & Wright (2009) are useful references for understanding the utility of random slope models.*

## Improving the accuracy of parameter estimation

Random effect models use data from all the groups to estimate the mean and variance of the global distribution of group means. Assuming all group means are drawn from a common distribution causes the estimates of their means to drift towards the global mean $\mu_{\text{group}}$. This phenomenon, known as *shrinkage* (Gelman & Hill, 2007; Kéry, 2010), can also lead to smaller and more precise standard errors around means. Shrinkage is strongest for groups with small sample sizes, as the paucity of within-group information to estimate the mean is counteracted by the model using data from other groups to improve the precision of the estimate. This 'partial pooling' of the estimates is a principal benefit of fitting something as a random effect (Gelman & Hill, 2007). However, it can feel strange that group means should be shrunk towards the global mean, especially for researchers more used to treating sample means as independent fixed effects. Accordingly, one issue is that variance estimates can be hugely imprecise when there are fewer than five levels of the random grouping variable (intercept or slope; see Harrison, 2015). However, thanks to the Central Limit Theorem, the assumption of Gaussian distribution of group means is usually a good one, and the benefits of hierarchical analysis will outweigh the apparent costs of shrinkage.

## Estimating variance components

In some cases, the variation among groups will be of interest to ecologists. For example, imagine we had measured the clutch masses of 30 individual birds, each of which had produced five clutches ($n = 150$). We might be interested in asking whether different females tend to produce consistently different clutch masses (high among-female variance for clutch mass). To do so, we might fit an intercept-only model with Clutch Mass as the response variable and a Gaussian error structure:

```
Model <- lmer(ClutchMass ~ 1 + (1|FemaleID)
```

By fitting individual 'FemaleID' as a random intercept term in the LMM, we estimate the among-female variance in our trait of interest. This model will also estimate the residual variance term, which we can use in conjunction with the among-female variance term to calculate an 'intra-class correlation coefficient' that measures individual repeatability in our trait (see Nakagawa & Schielzeth, 2010). While differences among individuals can be obtained by fitting individual ID as a fixed effect, this uses a degree of freedom for each individual ID after the first, severely limiting model power, and does not benefit from increased estimation accuracy through shrinkage. More importantly, repeatability scores derived from variance components analysis can be compared across studies for the same trait, and even across traits in the same study. Variance component analysis is a powerful tool for partitioning variation in a focal trait among biologically interesting groups, and several more complex examples exist
(see *Nakagawa & Schielzeth, 2010*; *Wilson et al., 2010*; *Houslay & Wilson, 2017*). In particular, quantitative genetic studies rely on variance component analysis for estimating the heritability of traits such as body mass or size of secondary sexual characteristics (*Wilson et al., 2010*). We recommend the tutorials in *Wilson et al. (2010)* and *Houslay & Wilson (2017)* for a deeper understanding of the power and flexibility of variance component analysis.

### Making predictions for unmeasured groups

Fixed effect estimates prevent us from making predictions for new groups because the model estimates are only relevant to groups in our dataset (*Zuur et al., 2009*, p. 327). Conversely, we can use the estimate of the global distribution of population means to predict for the average group using the mean of the distribution $\mu_{group}$ for a random effects model (see Fig. 1). We could also sample hypothetical groups from our random effect distribution, as we know its mean and SD (*Zuur & Ieno, 2016*). Therefore, whether something is fitted as a fixed or random effect can depend on the goal of the analysis: are we only interested in the mean values for each group in our dataset, or do we wish to use our results to extend our predictions to new groups? Even if we do not want to predict to new groups, we might wish to fit something as a random effect to take advantage of the shrinkage effect and improved parameter estimation accuracy.

## CONSIDERATIONS WHEN FITTING RANDOM EFFECTS

Random effect models have several desirable properties (see above), but their use comes with some caveats. First, they are quite 'data hungry'; requiring at least five 'levels' (groups) for a random intercept term to achieve robust estimates of variance (*Gelman & Hill, 2007*; *Harrison, 2015*). With <5 levels, the mixed model may not be able to estimate the among-population variance accurately. In this case, the variance estimate will either collapse to zero, making the model equivalent to an ordinary GLM (*Gelman & Hill, 2007*, p. 275) or be non-zero but incorrect if the small number of groups that were sampled are not representative of true distribution of means (*Harrison, 2015*). Second, models can be unstable if sample sizes across groups are highly unbalanced i.e. if some groups contain very few data. These issues are especially relevant to random slope models (*Grueber et al., 2011*). Third, an important issue is the difficulty in deciding the 'significance' or 'importance' of variance among groups. The variance of a random effect is inevitably at least zero, but how big does it need to be considered of interest? Fitting a factor as a fixed effect provides a statement of the significance of differences (variation) among groups relatively easily. Testing differences among levels of a random effect is made much more difficult for frequentist analyses, though not so in a Bayesian framework (*Kéry, 2010*, see '*Testing Significance of Random Effects*' section). Finally, an issue that is not often addressed is that of mis-specification of random effects. GLMMs are powerful tools, but incorrectly parameterising the random effects in the model could yield model estimates that are as unreliable as ignoring the need for random effects altogether. Examples include: (i) failure to recognise non-independence caused by nested structures in the data e.g. multiple clutch measures from a single bird; (ii) failing to specify random
slopes to prevent constraining slopes of predictors to be identical across clusters in the data (see *Barr et al., 2013*); and (iii) testing the significance of fixed effects at the wrong 'level' of hierarchical models that ultimately leads to pseudoreplication and inflated Type I error rates. Traditionally users of LMMs might have used *F*-tests of significance. *F*-tests are ill-advised for unbalanced experimental designs and irrelevant for non-Gaussian error structures, but they at least provide a check of model hierarchy using residual degrees of freedom for fixed effects. The now-standard use of likelihood ratio tests (LRT) of significance in LMMs means that users and readers have little opportunity to check the position of significance tests in the hierarchy of likelihoods.

*Further reading: Harrison (2015) shows how poor replication of the random intercept groups can give unstable model estimates. Zuur & Ieno (2016) discuss the importance of identifying dependency structures in the data.*

## DECIDING MODEL STRUCTURE FOR GLMMs

### Choosing error structures and link functions

General linear models make various statistical assumptions, including additivity of the linear predictors, independence of errors, equal variance of errors (homoscedasticity) and normality of errors (*Gelman & Hill, 2007*, p. 46; *Zuur et al., 2009*, p. 19). Ecologists often deal with response variables that violate these assumptions, and face several decisions about model specification to ensure models of such data are robust. The price for ignoring violation of these assumptions tends to be an inflated Type I error rate (*Zuur, Ieno & Elphick, 2010*; *Ives, 2015*). In some cases, however, transformation of the response variable may be required to ensure these assumptions are met. For example, an analytical goal may be to quantify differences in mean mass between males and females, but if the variance in mass for one sex is greater than the other, the assumption of homogeneity of variance is violated. Transformation of the data can remedy this (*Zuur et al., 2009*); 'mean-variance stabilising transformations' aim to make the variance around the fitted mean of each group homogenous, making the models more robust. Alternatively, modern statistical tools such as the 'varIdent' function in the R package *nlme* can allow one to explicitly model differences in variance between groups to avoid the need for data transformation.

*Further reading: Zuur, Ieno & Elphick (2010) provide a comprehensive guide on using data exploration techniques to check model assumptions, and give advice on transformations.*

For non-Gaussian data, our modelling choices become more complex. Non-Gaussian data structures include, for example, Poisson-distributed counts (number of eggs laid, number of parasites); binomial-distributed constrained counts (number of eggs that hatched in a clutch; prevalence of parasitic infection in a group of hosts) or Bernoulli-distributed binary traits (e.g. infected with a parasite or not). Gaussian models of these data would violate the assumptions of normality of errors and homogenous variance. To model these data, we have two initial choices: (i) we can apply a transformation to our non-Gaussian response to 'make it' approximately Gaussian, and then use a Gaussian model; or (ii) we can apply a GL(M)M and specify the appropriate error distribution and link function. The link function takes into account the (assumed) empirical

distribution of our data by transformation of the linear predictor within the model. It is critical to note that transformation of the raw response variable is not equivalent to using a link function to apply a transformation in the model. Data-transformation applies the transformation to the raw response, whilst using a link function transforms the fitted mean (the linear predictor). That is, *the mean of a log-transformed response (using a data transformation) is not identical to the logarithm of a fitted mean (using a link function).*

The issue of transforming non-Gaussian data to fit Gaussian models to them is contentious. For example, arcsin square-root transformation of proportion data was once extremely common, but recent work has shown it to be unreliable at detecting real effects (*Warton & Hui, 2011*). Both logit-transformation (for proportional data) and Binomial GLMMs (for binary response variables) have been shown to be more robust (*Warton & Hui, 2011*). *O'Hara & Kotze (2010)* argued that log-transformation of count data performed well in only a small number of circumstances (low dispersion, high mean counts), which are unlikely to be applicable to ecological datasets. However, *Ives (2015)* recently countered these assumptions with evidence that transformed count data analysed using LMMs can often outperform Poisson GLMMs. We do not make a case for either here, but acknowledge the fact that there is unlikely to be a universally best approach; each method will have its own strengths and weakness depending on the properties of the data (*O'Hara & Kotze, 2010*). Checking the assumptions of the LMM or GLMM is an essential step (see section Quantifying GLMM Fit and Performance). An issue with transformations of non-Gaussian data is having to deal with zeroes as special cases (e.g. you can't log transform a 0), so researchers often add a small constant to all data to make the transformation work, a practice that has been criticised (*O'Hara & Kotze, 2010*). GLMMs remove the need for these 'adjustments' of the data. The important point here is that transformations change the entire relationship between *Y* and *X* (*Zuur et al., 2009*), but different transformations do this to different extents and it may be impossible to know which transformation is best without performing simulations to test the efficacy of each (*Warton & Hui, 2011*; *Ives, 2015*).

*Further reading: Crawley (2013, Ch 13) gives a broad introduction to the various error structures and link functions available in the R statistical framework. O'Hara & Kotze (2010); Ives (2015) and Warton et al. (2016) argue the relative merits of GLMs vs log-transformation of count data; Warton & Hui (2011) address the utility of logit-transformation of proportion data compared to arcsin square-root transformation.*

## Choosing random effects I: crossed or nested?

A common issue that causes confusion is this issue of specifying random effects as either 'crossed' or 'nested'. In reality, the way you specify your random effects will be determined by your experimental or sampling design (*Schielzeth & Nakagawa, 2013*). A simple example can illustrate the difference. Imagine a researcher was interested in understanding the factors affecting the clutch mass of a passerine bird. They have a study population spread across five separate woodlands, each containing 30 nest boxes. Every week during breeding they measure the foraging rate of females at feeders, and measure their subsequent clutch mass. Some females have multiple clutches in a season and contribute

multiple data points. Here, female ID is said to be *nested within woodland*: each woodland contains multiple females unique to that woodland (that never move among woodlands). The nested random effect controls for the fact that (i) clutches from the same female are not independent, and (ii) females from the same woodland may have clutch masses more similar to one another than to females from other woodlands

```
Clutch Mass ~ Foraging Rate + (1|Woodland/Female ID)
```

Now imagine that this is a long-term study, and the researcher returns every year for five years to continue with measurements. Here it is appropriate fit year as a *crossed* random effect because every woodland appears multiple times in every year of the dataset, and females that survive from one year to the next will also appear in multiple years.

```
Clutch Mass ~ Foraging Rate + (1|Woodland/Female ID)+ (1|Year)
```

Understanding whether your experimental/sampling design calls for nested or crossed random effects is not always straightforward, but it can help to visualise experimental design by drawing it (see *Schielzeth & Nakagawa, 2013*; Fig. 1), or tabulating your observations by these grouping factors (e.g. with the '*table*' command in R) to identify how your data are distributed. We advocate that researchers always ensure that their levels of random effect grouping variables are uniquely labelled. For example, females are labelled $1-n$ in each woodland, the model will try and pool variance for all females with the same code. Giving all females a unique code makes the nested structure of the data is implicit, and a model specified as ~(1| Woodland) + (1|FemaleID) would be identical to the model above.

Finally, we caution that whether two factors are nested or crossed affects the ability of (G)LMMs to estimate the effect of the interaction between those two factors on the outcome variable. Crossed factors allow the model to accurately estimate the interaction effects between the two, whereas nested factors automatically pool those effects in the second (nested) factor (*Schielzeth & Nakagawa, 2013*). We do not expand on this important issue here but direct the reader to *Schielzeth & Nakagawa (2013)* for an excellent treatment of the topic.

## Choosing random effects II: random slopes

Fitting random slope models in ecology is not very common. Often, researchers fit random intercepts to control for non-independence among measurements of a statistical group (e.g. birds within a woodland), but force variables to have a common slope across all experimental units. However, there is growing evidence that researchers should be fitting random slopes as standard practice in (G)LMMs. Random slope models allow the coefficient of a predictor to vary based on clustering/non-independence in the data (see Fig. 1B). In our bird example above, we might fit a random slope for the effect of foraging rate on clutch mass given each individual bird ID. That is, the magnitude of the effect foraging rate on resultant clutch mass differs among birds. Random slope models (also often called random coefficients models, *Kéry, 2010*) apply to both continuous and factor variables. For example, if we had applied a two-level feeding
treatment to birds in each woodland (vitamin supplementation or control), we might also expect the magnitude of the effect of receiving vitamin supplementation to differ depending on which woodland it was applied to. So here we would specify random slopes for the treatment variable given woodland ID.

*Schielzeth & Forstmeier (2009)* found that including random slopes controls Type I error rate (yields more accurate *p* values), but also gives more power to detect among individual variation. *Barr et al. (2013)* suggest that researchers should fit the maximal random effects structure possible for the data. That is, if there are four predictors under consideration, all four should be allowed to have random slopes. However, we believe this is unrealistic because random slope models require large numbers of data to estimate variances and covariances accurately (*Bates et al., 2015a*). Ecological datasets can often struggle to estimate a single random slope, diagnosed by a perfect correlation (1 or −1) between random intercepts and slopes (*Bates et al., 2015a*). Therefore, the approach of fitting the 'maximal' complexity of random effects structure (*Barr et al., 2013*) is perhaps better phrased as fitting the most complex mixed effects structure allowed by your data (*Bates et al., 2015a*), which may mean either (i) fitting random slopes but removing the correlation between intercepts and slopes; or (ii) fitting no random slopes at all but accepting that this likely inflates the Type I error rate (*Schielzeth & Forstmeier, 2009*). If fitting a random slope model including correlations between intercepts and slopes, always inspect the intercept-slope correlation coefficient in the variance/covariance summary returned by packages like *lme4* to look for evidence of perfect correlations, indicative of insufficient data to estimate the model.

*Further Reading: Schielzeth & Forstmeier (2009) is essential reading for understanding how random slopes control Type I error rate, and Bates et al. (2015a) gives sound advice on how to iteratively determine optimal complexity of random effect structure. Barr et al. (2013) and Aarts et al. (2015) discuss the merits of fitting random slopes to clustered data to control false positive rates.*

## Choosing fixed effect predictors and interactions

One of the most important decisions during the modelling process is deciding which predictors and interactions to include in models. Best practice demands that each model should represent a specific a priori hypothesis concerning the drivers of patterns in data (*Burnham & Anderson, 2002*; *Forstmeier & Schielzeth, 2011*), allowing the assessment of the relative support for these hypotheses in the data irrespective of model selection philosophy. The definition of 'hypothesis' must be broadened from the strict pairing of null and alternative that is classically drilled into young pupils of statistics and experimental design. Frequentist approaches to statistical modelling still work with nested pairs of hypotheses. Information theorists work with whole sets of competing hypotheses. Bayesian modellers are comfortable with the idea that every possible parameter estimate is a hypothesis in its own right. But these epistemological differences do not really help to solve the problem of 'which' predictors should be considered valid members of the full set to be used in a statistical modelling exercise. It is therefore often unclear how best to design the most complex model, often referred to

as the *maximal model* (which contains all factors, interactions and covariates that might be of any interest, *Crawley, 2013*) or as the *global model* (a highly parameterized model containing the variables and associated parameters thought to be important of the problem at hand, *Burnham & Anderson, 2002*; *Grueber et al., 2011*). We shall use the latter term here for consistency with terminology used in information-theory (*Grueber et al., 2011*).

Deciding which terms to include in the model requires careful and rigorous a priori consideration of the system under study. This may appear obvious; however diverse authors have noticed a lack of careful thinking when selecting variables for inclusion in a model (*Peters, 1991*; *Chatfield, 1995*; *Burnham & Anderson, 2002*). Lack of a priori consideration, of what models represent, distinguishes rigorous hypothesis testing from 'fishing expeditions' that seek significant predictors among a large group of contenders. Ideally, the global model should be carefully constructed using the researchers' knowledge and understanding of the system such that only predictors likely to be pertinent to the problem at hand are included, rather than including all the predictors the researcher has collected and/or has available. This is a pertinent issue in the age of 'big data', where researchers are often overwhelmed with predictors and risk skipping the important step of a priori hypothesis design. In practice, for peer reviewers it is easy to distinguish fishing expeditions from a priori hypothesis sets based on the evidence base presented in introductory sections of research outputs.

## How complex should my global model be?

The complexity of the global model will likely be a trade-off between the number of measured observations (the $n$ of the study) and the proposed hypotheses about how the measured variables affect the outcome (response) variable. Lack of careful consideration of the parameters to be estimated can result in overparameterised models, where there are insufficient data to estimate coefficients robustly (*Southwood & Henderson, 2000*; *Quinn & Keough, 2002*; *Crawley, 2013*). In simple GLMs, overparameterisation results in a rapid decline in (or absence of) degrees of freedom with which to estimate residual error. Detection of overparameterisation in LMMs can be more difficult because each random effect uses only a single degree of freedom, however the estimation of variance among small numbers of groups can be numerically unstable. Unfortunately, it is common practice to fit a global model that is simply as complex as possible, irrespective of what that model actually represents; that is a dataset containing $k$ predictors yields a model containing a $k$-way interaction among all predictors and simplify from there (*Crawley, 2013*). This approach is flawed for two reasons. First, this practice encourages fitting biologically-unfeasible models containing nonsensical interactions. It should be possible to draw and/or visualise what the fitted model 'looks like' for various combinations of predictors—generally extremely difficult when more than two terms are interacting. Second, using this approach makes it very easy to fit a model too complex for the data. At best, the model will fail to converge, thus preventing inference. At worst, the model will 'work', risking false inference. Guidelines for the ideal ratio of data points ($n$) to estimated parameters ($k$) vary widely (see *Forstmeier & Schielzeth, 2011*). *Crawley (2013)* suggests a minimum $n/k$ of 3,

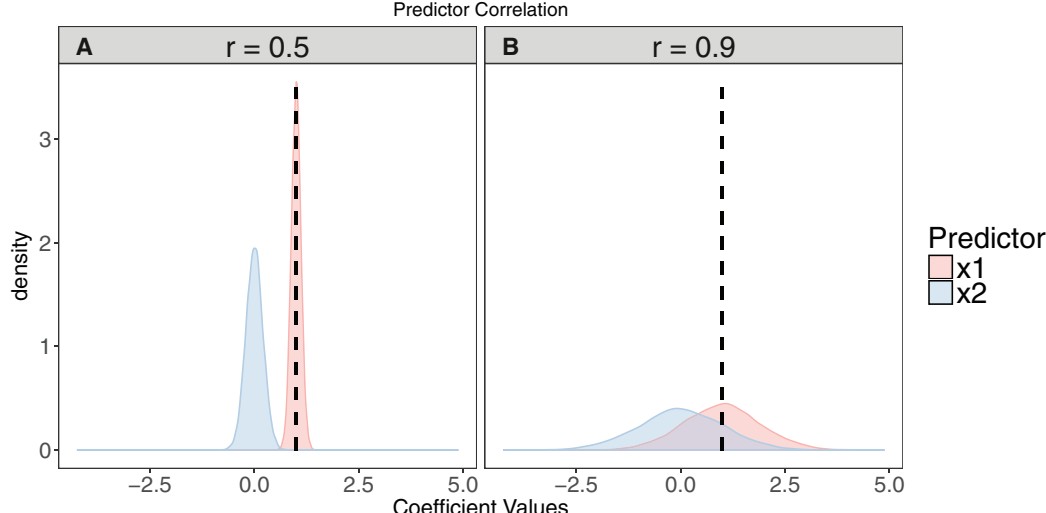

**Figure 2** **The effect of collinearity on model parameter estimates.** We simulated 10,000 iterations of a model $y \sim x1 + x2$, where $x1$ had a positive effect on $y$ ($\beta_{x1} = 1$, vertical dashed line). $x2$ is collinear with $x1$ with either a moderate ($r = 0.5$). (A) or strong correlation ($r = 0.9$). (B) With moderate collinearity, estimation of $\beta_{x1}$ is precise, but certainty of the sign of $\beta_{x2}$ is low. When collinearity is strong, estimation of $\beta_{x1}$ is far less precise, with 14% of simulations estimating a negative coefficient for the effect of $x1$. For more elaborate versions of these simulations, see *Freckleton (2011)*.

though we argue this is very low and that an *n/k* of 10 is more conservative. A 'simple' model containing a three-way interaction between continuous predictors, all that interaction's daughter terms, and a single random intercept needs to estimate eight parameters, so requires a dataset of a *minimum n* of 80 using this rule. Interactions can be especially demanding, as fitting interactions between a multi-level factor and a continuous predictor can result in poor sample sizes for specific treatment combinations even if the total *n* is quite large (*Zuur, Ieno & Elphick, 2010*), which will lead to unreliable model estimates.

*Grueber et al. (2011)* show an excellent worked example of a case where the most complex model is biologically feasible and well-reasoned, containing only one two-way interaction. *Nakagawa & Foster (2004)* discuss the use of power analyses, which will be useful in determining the appropriate n/k ratio for a given system.

### Assessing predictor collinearity

With the desired set of predictors identified, it is wise to check for collinearity among predictor variables. Collinearity among predictors can cause several problems in model interpretation because those predictors explain some of the same variance in the response variable, and their effects cannot be estimated independently (*Quinn & Keough, 2002*; *Graham, 2003*): First, it can cause model convergence issues as models struggle to partition variance between predictor variables. Second, positively correlated variables can have negatively correlated regression coefficients, as the marginal effect of one is estimated, given the effect of the other, leading to incorrect interpretations of the direction of effects (Fig. 2). Third, collinearity can inflate standard errors of coefficient estimates

and make 'true' effects harder to detect (*Zuur, Ieno & Elphick, 2010*). Finally, collinearity can affect the accuracy of model averaged parameter estimates during multi-model inference (*Freckleton, 2011*; *Cade, 2015*). Examples of collinear variables include climatic data such as temperature and rainfall, and morphometric data such as body length and mass. Collinearity can be detected in several ways, including creating correlation matrices between raw explanatory variables, with values >0.7 suggesting both should not be used in the same model (*Dormann et al., 2013*); or calculating the variance inflation factor (VIF) of each predictor that is a candidate for inclusion in a model (details in *Zuur, Ieno & Elphick, 2010*) and dropping variables with a VIF higher than a certain value (e.g. 3, *Zuur, Ieno & Elphick, 2010*; or 10, *Quinn & Keough, 2002*). One problem with these methods though is that they rely on a user-selected choice of threshold of either the correlation coefficient or the VIF, and use of more stringent (lower) is probably sensible. Some argue that one should always prefer inspection of VIF values over correlation coefficients of raw predictors because strong multicollinearity can be hard to detect with the latter. When collinearity is detected, researchers can either select one variable as representative of multiple collinear variables (*Austin, 2002*), ideally using biological knowledge/reasoning to select the most meaningful variable (*Zuur, Ieno & Elphick, 2010*); or conduct a dimension-reduction analysis (e.g. Principal Components Analysis; *James & McCullugh, 1990*), leaving a single variable that accounts for most of the shared variance among the correlated variables. Both approaches will only be applicable if it is possible to group explanatory variables by common features, thereby effectively creating broader, but still meaningful explanatory categories. For instance, by using mass and body length metrics to create a 'scaled mass index' representative of body size (*Peig & Green, 2009*).

### Standardising and centring predictors

Transformations of predictor variables are common, and can improve model performance and interpretability (*Gelman & Hill, 2007*). Two common transformations for continuous predictors are (i) predictor centering, the mean of predictor $x$ is subtracted from every value in $x$, giving a variable with mean 0 and SD on the original scale of $x$; and (ii) predictor standardising, where $x$ is centred and then divided by the SD of $x$, giving a variable with mean 0 and SD 1. Rescaling the mean of predictors containing large values (e.g. rainfall measured in 1,000s of millimetre) through centring/standardising will often solve convergence problems, in part because the estimation of intercepts is brought into the main body of the data themselves. Both approaches also remove the correlation between main effects and their interactions, making main effects more easily interpretable when models also contain interactions (*Schielzeth, 2010*). Note that this collinearity among coefficients is distinct from collinearity between two separate predictors (see above). Centring and standardising by the mean of a variable changes the interpretation of the model intercept to the value of the outcome expected when $x$ is at its mean value. Standardising further adjusts the interpretation of the coefficient (slope) for $x$ in the model to the change in the outcome variable for a 1 SD change in the value of $x$. Scaling is therefore a useful tool to improve the stability of models and

likelihood of model convergence, and the accuracy of parameter estimates *if* variables in a model are on large (e.g. 1,000s of millimetre of rainfall), or vastly different scales. When using scaling, care must be taken in the interpretation and graphical representation of outcomes.

*Further reading: Schielzeth (2010) provides an excellent reference to the advantages of centring and standardising predictors. Gelman (2008) provides strong arguments for standardising continuous variables by 2 SDs when binary predictors are in the model. Gelman & Hill (2007, p. 56, 434) discuss the utility of centring by values other than the mean.*

## Quantifying GLMM fit and performance

Once a global model is specified, it is vital to quantify model fit and report these metrics in the manuscript. The global model is considered the best candidate for assessing fit statistics such as overdispersion (*Burnham & Anderson, 2002*). Information criteria scores should not be used as a proxy for model fit, because a large difference in AIC between the top and null models is not evidence of a good fit. AIC tells us nothing about whether the basic distributional and structural assumptions of the model have been violated. Similarly, a high $R^2$ value is in itself only a test of the magnitude of model fit and not an adequate surrogate for proper model checks. Just because a model has a high $R^2$ value does not mean it will pass checks for assumptions such as homogeneity of variance. We strongly encourage researchers to view *model fit* and *model adequacy* as two separate but equally important traits that must be assessed and reported. Model fit can be poor for several reasons, including the presence of overdispersion, failing to include interactions among predictors, failing to account for non-linear effects of the predictors on the response, or specifying a sub-optimal error structure and/or link function. Here we discuss some key metrics of fit and adequacy that should be considered.

### *Inspection of residuals and linear model assumptions*

Best practice is to examine plots of residuals versus fitted values for the entire model, as well as model residuals versus all explanatory variables to look for patterns (*Zuur, Ieno & Elphick, 2010*; *Zuur & Ieno, 2016*). In addition, there are further model checks specific to mixed models. First, inspect residuals versus fitted values for each grouping level of a random intercept factor (*Zuur et al., 2009*). This will often prove dissatisfying if there are few data/residuals per group, however this in itself is a warning flag that the assumptions of the model might be based on weak foundations. Note that, for GLMMs, it is wise to use normalised/Pearson residual when looking for patterns, as they account for the mean-variance relationship of generalized models (*Zuur et al., 2009*). Another feature of fit that is very rarely tested for in (G)LMMs is the assumption of normality of deviations of the conditional means of the random effects from the global intercept. Just as a quantile–quantile (QQ) plot of linear model residuals should show points falling along a straight line (*Crawley, 2013*), so should a QQ plot of the random effect means (*Schielzeth & Nakagawa, 2013*).

*Further reading: Zuur, Ieno & Elphick (2010) give an excellent overview of the assumptions of linear models and how to test for their violation. See also Gelman & Hill (2007, p. 45).*

*The R package 'sjPlot' (Lüdecke, 2017) has built in functions for several LMM diagnostics, including random effect QQ plots. Zuur et al. (2009) provides a vast selection of model diagnostic techniques for a host of model types, including Generalised Least Squared (GLS), GLMMs and Generalized Additive Mixed Effects Models.*

### Overdispersion

Models with a Gaussian (Normal) error structure do not require adjustment for overdispersion, as Gaussian models do not assume a specific mean-variance relationship. For generalized mixed models (GLMMs), however (e.g. Poisson, Binomial), the variance of the data can be greater than predicted by the error structure of the model (*Hilbe, 2011*). Overdispersion can be caused by several processes influencing data, including zero-inflation, aggregation (non-independence) among counts, or both (*Zuur et al., 2009*). The presence of overdispersion in a model suggests it is a bad fit, and standard errors of estimates will likely be biased unless overdispersion is accounted for (*Harrison, 2014*). The use of canonical binomial and Poisson error structures, when residuals are overdispersed, tends to result in Type I errors because standard errors are underestimated. Adding an observation-level random effect (OLRE) to overdispersed Poisson or Binomial models can model the overdispersion and give more accurate estimates of standard errors (*Harrison, 2014, 2015*). However, OLRE models may yield inferior fit and/or biased parameter estimates compared to models using compound probability distributions such as the Negative-Binomial for count data (*Hilbe, 2011*; *Harrison, 2014*) or Beta-Binomial for proportion data (*Harrison, 2015*), and so it is good practice to assess the relative fit of both types of model using AIC before proceeding (*Zuur et al., 2009*). Researchers very rarely report the overdispersion statistic (but see *Elston et al., 2001*), and it should be made a matter of routine. See "Assessing Model Fit Through Simulation" Section for advice on how to quantify and model overdispersion. Note that models can also be underdispersed (*less* variance than expected/predicted by the model, but the tools for dealing with underdispersion are less well developed (*Zuur et al., 2009*). The *spaMM* package (*Rousset & Ferdy, 2014*) can fit models that can handle both overdispersion and underdispersion.

*Further reading: Crawley (2013, pp. 580–581) gives an elegant demonstration of how failing to account for overdispersion leads to artificially small standard errors and spurious significance of variables. Harrison (2014) quantifies the ability of OLRE to cope with overdispersion in Poisson models. Harrison (2015) compares Beta-Binomial and OLRE models for overdispersed proportion data.*

### $R^2$

In a linear modelling context, $R^2$ gives a measure of the proportion of explained variance in the model, and is an intuitive metric for assessing model fit. Unfortunately, the issue of calculating $R^2$ for (G)LMMs is particularly contentious; whereas residual variance can easily be estimated for a simple linear model with no random effects and a Normal error structure, this is not the case for (G)LMMS. In fact, two issues exist with generalising $R^2$ measures to (G)LMMs: (i) for generalised models containing non-Normal error

structures, it is not clear how to calculate the residual variance term on which the $R^2$ term is dependent; and (ii) for mixed effects models, which are hierarchical in nature and contain error (unexplained variance) at each of these levels, it is uncertain which level to use to calculate a residual error term (*Nakagawa & Schielzeth, 2013*). Diverse methods have been proposed to account for this in GLMMs, including multiple so-called 'pseudo-$r^2$' measures of explained variance (*Nagelkerke, 1991*; *Cox & Snell, 1989*), but their performance is often unstable for mixed models and can return negative values (*Nakagawa & Schielzeth, 2013*). *Gelman & Pardoe (2006)* derived a measure of $R^2$ that accounts for the hierarchical nature of LMMs and gives a measure for both group and unit level regressions (see also *Gelman & Hill, 2007*, p. 474), but it was developed for a Bayesian framework and a frequentist analogue does not appear to be widely implemented. The method that has gained the most support over recent years is that of *Nakagawa & Schielzeth (2013)*.

The strength of the *Nakagawa & Schielzeth (2013)* method for GLMMs is that it returns two complementary $R^2$ values: the marginal $R^2$ encompassing variance explained by only the fixed effects, and the conditional $R^2$ comprising variance explained by both fixed and random effects i.e. the variance explained by the whole model (*Nakagawa & Schielzeth, 2013*). Ideally, both should be reported in publications as they provide different information; which one is more 'useful' may depend on the rationale for specifying random effects in the first instance. Recently, *Nakagawa, Johnson & Schielzeth (2017)* expanded their $R^2$ method to handle models with compound probability distributions like the Negative Binomial error family. Note that when OLREs are included (see 'Overdispersion' section above), the conditional $R^2$ becomes less useful as a measure of explained variance because it includes the extra-parametric dispersion being modelled, but has no predictive power (*Harrison, 2014*).

*Further reading: Nakagawa & Schielzeth (2013) provide an excellent and accessible description of the problems with, and solutions to, generalising $R^2$ metrics to GLMMs. The Nakagawa & Schielzeth (2013) $R^2$ functions have been incorporated into several packages, including 'MuMIn' (Bartoń, 2016) and 'piecewiseSEM' (Lefcheck, 2015), and Johnson (2014) has developed an extension of the functions for random slope models. See Harrison (2014) for a cautionary tale of how the GLMM $R^2$ functions are artificially inflated for overdispersed models.*

### Stability of variance components and testing significance of random effects

When models are too complex relative to the amount of data available, GLMM variance estimates can collapse to zero (they cannot be negative, not to be confused with *co*variance estimates which can be negative). This is not a problem per se, but it's important to acknowledge that in this case the model is equivalent to a standard GLM. Reducing model complexity by removing interactions will often allow random effects variance component estimates to become >0, but this is problematic if quantifying the interaction is the primary goal of the study. REML (restricted/residual maximum likelihood) should be used for estimating variance components of random effects in Gaussian models as it

produces less biased estimates compared to maximum likelihood (ML) (*Bolker et al., 2009*). However, when comparing two models with the same random structure but different fixed effects, ML estimation cannot easily be avoided. The RLRsim package (*Scheipl & Bolker, 2016*) can be used to calculate restricted LTRs for variance components in mixed and additive models. Crucially, when testing the significance of a variance component we are 'testing on the boundary' (*Bolker et al., 2009*). That is the null hypothesis for random effects ($\sigma = 0$) is at the boundary of its possible range (it has to be $\geq 0$), meaning $p$-values from a LRT are inaccurate. Dividing $p$ values by two for tests of single variance components provides an approximation to remedy this problem (*Verbenke & Molenberghs, 2000*).

Finally, estimating degrees of freedom for tests of random effects is difficult, as a random effect can theoretically use anywhere between 1 and $N − 1$ d$f$ (where $N$ is the number of random-effect levels) (*Bolker et al., 2009*). Adequate F and $P$ values can be calculated using Satterthwaite or Kenward–Roger approximations to determine denominator degrees of freedom, the former being implemented in the package 'lmerTest' (*Kuznetsova, Brockhoff & Christensen, 2014*, see further details in section Model Selection and Multi-Model Inference below).

### Assessing model fit through simulation

Simulation is a powerful tool for assessing model fit (*Gelman & Hill, 2007*; *Kéry, 2010*; *Zuur & Ieno, 2016*), but is rarely used. The premise here is simple: when simulating a response variable from a given set of parameter estimates (a model), the fit of the model to those *simulated* 'ideal' response data should be comparable to the model's fit to the real response variable (*Kéry, 2010*). Each iteration yields a simulated dataset that allows calculation of a statistic of interest such as the sum of squared residuals (*Kéry, 2010*), the overdispersion statistic (*Harrison, 2014*) or the percentage of zeroes for a Poisson model (*Zuur & Ieno, 2016*). If the model is a good fit, after a sufficiently large number of iterations (e.g. 10,000) the distribution of this statistic should encompass the observed statistic in the real data. Significant deviations outside of that distribution indicate the model is a poor fit (*Kéry, 2010*). Figure 3 shows an example of using simulation to assess the fit of a Poisson GLMM. After fitting a GLMM to count data, we may wish to check for overdispersion and/or zero-inflation, the presence of which might suggest we need to adjust our modelling strategy. Simulating 10,000 datasets from our model reveals that the proportion of zeroes in our real data is comparable to simulated expectation (Fig. 3A). Conversely, simulating 1,000 datasets and refitting our model to each dataset, we see that the sum of the squared Pearson residuals for the real data is far larger than simulated expectation (Fig. 3B), giving evidence of overdispersion (*Harrison, 2014*). We can use the simulated frequency distribution of this test statistic to derive a mean and 95% confidence interval for the overdispersion by calculating the ratio of our test statistic to the simulated values (*Harrison, 2014*). The dispersion statistic for our model is 3.16 (95% CI [2.77–3.59]). Thus, simulations have allowed us to conclude that our model is overdispersed, but that this overdispersion is not due to zero-inflation. All R code for reproducing these simulations is provided in the Figshare: DOI 10.6084/m9.figshare.5173396.

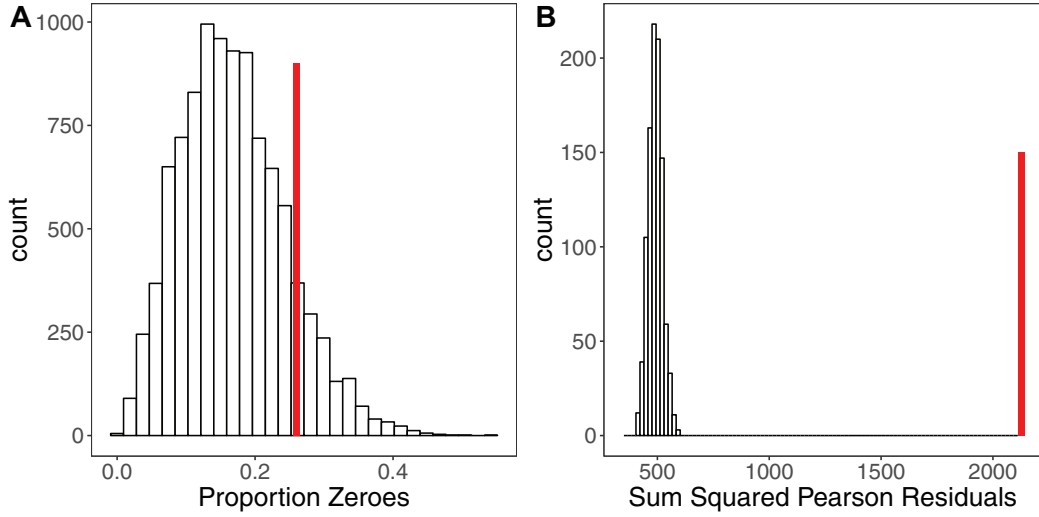

**Figure 3  Using Simulation to Assess Model Fit for GLMMs.** (A) Histogram of the proportion of zeroes in 10,000 datasets simulated from a Poisson GLMM. Vertical red line shows the proportion of zeroes in our real dataset. There is no strong evidence of zero-inflation for these data. (B) Histogram of the sum of squared Pearson residuals for 1,000 parametric bootstraps where the Poisson GLMM has been re-fitted to the data at each step. Vertical red line shows the test statistic for the original model, which lies well outside the simulated frequency distribution. The ratio of the real statistic to the simulated data can be used to calculate a mean dispersion statistic and 95% confidence intervals, which for these data is mean 3.16, (95% CI [2.77–3.59]). Simulating from models provides a simple yet powerful set of tools for assessing model fit and robustness.            

*Further reading: The R package 'SQuiD' (Allegue et al., 2017) provides a highly flexible simulation tool for learning about, and exploring the performance of, GLMMs. Rykiel (1996) discusses the need for validation of models in ecology.*

### Dealing with missing data

When collecting ecological data it is often not possible to measure all of the predictors of interest for every measurement of the dependant variable. Such missing data are a common feature of ecological datasets, however the impacts of this have seldom been considered in the literature (Nakagawa & Freckleton, 2011). Incomplete rows of data in dataframes i.e. those missing predictor and/or response variables are often dealt with by deleting or ignoring those rows of data entirely when modelling (Nakagawa & Freckleton, 2008), although this may result in biased parameter estimates and, depending on the mechanism underlying the missing data, reduces statistical power (Nakagawa & Freckleton, 2008). Nakagawa & Freckleton (2011) recommend multiple imputation (MI) as a mechanism for handling non-informative missing data, and highlight the ability of this technique for more accurate estimates, particularly for IT/AIC approaches.

*Further reading: See Nakagawa & Freckleton (2008) for a review on the risks of ignoring incomplete data. Nakagawa & Freckleton (2011) demonstrate the effects of missing data during model selection procedures, and provide an overview of R packages available for MI. Nakagawa (2015) and Noble & Nakagawa (2017) discuss methods for dealing with missing data in ecological statistics.*

## MODEL SELECTION AND MULTI-MODEL INFERENCE

Model selection seeks to optimise the trade-off between the fit of a model given the data and that model's complexity. Given that the researcher has a robust global model that satisfies standard assumptions of error structure and hierarchical independence, several methods of model selection are available, each of which maximises the fit-complexity trade off in a different way (*Johnson & Omland, 2004*). We discuss the relative merits of each approach briefly here, before expanding on the use of information-theory and multi-model inference in ecology. We note that these discussions are not meant to be exhaustive comparisons, and we encourage the reader to delve into the references provided for a comprehensive picture of the arguments for and against each approach.

### Stepwise selection, LRTs and *p* values

A common approach to model selection is the comparison of a candidate model containing a term of interest to the corresponding 'null' model lacking that term, using a *p* value from a LRT, referred to as null-hypothesis significance testing (NHST; *Nickerson, 2000*). Stepwise deletion is a model selection technique that drops terms sequentially from the global model to arrive at a 'minimal adequate model' (MAM). Evaluating whether a term should be dropped or not can be done using NHST to arrive at a model containing only significant predictors (see *Crawley, 2013*), or using IT to yield a model containing only terms that cause large increases in information criterion score when removed. Stepwise selection using NHST is by far the most common variant of this approach, and so we focus on this method here.

Stepwise deletion procedures have come under heavy criticism; they can overestimate the effect size of significant predictors (*Whittingham et al., 2006*; *Forstmeier & Schielzeth, 2011*; *Burnham, Anderson & Huyvaert, 2011*) and force the researcher to focus on a single best model as if it were the only combination of predictors with support in the data. Although we strive for simplicity and parsimony, this assumption is not always reasonable in complex ecological systems (*Burnham, Anderson & Huyvaert, 2011*). It is common to present the MAM as if it arose from a single a priori hypothesis, when in fact arriving at the MAM required multiple significance tests (*Whittingham et al., 2006*; *Forstmeier & Schielzeth, 2011*). This cryptic multiple testing can lead to hugely inflated Type I errors (*Forstmeier & Schielzeth, 2011*). Perhaps most importantly, LRT can be unreliable for fixed effects in GLMMs unless both total sample size and replication of the random effect terms is high (see *Bolker et al., 2009* and references therein), conditions which are often not satisfied for most ecological datasets. Because stepwise deletion can cause biased effect sizes, presenting means and SEs of parameters from the global model should be more robust, especially when the *n/k* ratio is low (*Forstmeier & Schielzeth, 2011*). Performing 'full model tests' (comparing the global model to an intercept only model) before investigating single-predictor effects controls the Type I error rate (*Forstmeier & Schielzeth, 2011*). Reporting the full model also helps reduce publication bias towards strong effects, providing future meta-analyses with estimates of both significant and non-significant effects (*Forstmeier & Schielzeth, 2011*). Global model reporting should

not replace other model selection methods, but provides a robust measure of how likely significant effects are to arise by sampling variation alone.

*Further reading: See Murtaugh (2009), and Murtaugh's (2014) excellent 'in Defense of p values', as well as the other papers on the topic in the same special issue of Ecology. Stephens et al. (2005) & Mundry (2011) argue the case for NHST under certain circumstances such as well-designed experiments. Halsey et al. (2015) discuss the wider issues of the reliability of p values relative to sample size.*

## Information-theory and multi-model inference

Unlike NHST, which leads to a focus on a single best model, model selection using IT approaches allows the degree of support in the data for several competing models to be ranked using metrics such as AIC. Information criteria attempt to quantify the Kullback–Leibler distance (KLD), a measure of the relative amount of information lost when a given model approximates the true data-generating process. Thus, relative difference among models in AIC should be representative in relative differences in KLD, and the model with the lowest AIC should lose the least information and be the best model in that it optimises the trade-off between fit and complexity (*Richards, 2008*). A key strength of the IT approach is that it accounts for 'model selection uncertainty', the idea that several competing models may all fit the data similarly well (*Burnham & Anderson, 2002*; *Burnham, Anderson & Huyvaert, 2011*). This is particularly useful when competing models share equal 'complexity' (i.e. number of predictors, or number of residual degrees of freedom): in such situations, NHST is impossible because NHST requires a simpler (nested) model for comparison. Where several models have similar support in the data, inference can be made from all models using model-averaging (*Burnham & Anderson, 2002*; *Johnson & Omland, 2004*; *Grueber et al., 2011*). Model averaging incorporates uncertainty by weighting the parameter estimate of a model by that model's Akaike weight (often referred to as the probability of that model being the best Kullback–Leibler model given the data, but see *Richards, 2005*). Multi-model inference places a strong emphasis on a priori formulation of hypotheses (*Burnham & Anderson, 2002*; *Dochtermann & Jenkins, 2011*; *Lindberg, Schmidt & Walker, 2015*), and model-averaged parameter estimates arising from multi-model inference are thought to lead to more robust conclusions about the biological systems compared to NHST (*Johnson & Omland, 2004*, but see *Richards, Whittingham & Stephens, 2011*). These strengths over NHST have meant that the use of IT approaches in ecology and evolution has grown rapidly in recent years (*Lindberg, Schmidt & Walker, 2015*; *Barker & Link, 2015*; *Cade, 2015*). We do not expand on the specific details of the difference between NHST and IT here, but point the reader to some excellent references on the topic. Instead, we use this section to highlight recent empirical developments in the best practice methods for the application of IT in ecology and evolution.

*Further reading: Grueber et al. (2011) and Symonds & Moussalli (2011) give a broad overview of multi-model inference in ecology, and provide a worked model selection exercise. Hegyi & Garamszegi (2011) provide a detailed comparison of IT and NHST*

*approaches. Burnham, Anderson & Huyvaert (2011) demonstrate how AIC approximates Kullback–Leibler information and provide some excellent guides for the best practice of applying IT methods to biological datasets. Vaida & Blanchard (2005) provide details on how AIC should be implemented for the analysis of clustered data.*

## PRACTICAL ISSUES WITH APPLYING INFORMATION THEORY TO BIOLOGICAL DATA

### Using all-subsets selection

All-Subsets selection is the act of fitting a global model, often containing every possible interaction, and then fitting every possible nested model. On the surface, all-subsets might appear to be a convenient and fast way of 'uncovering' the causal relationships in the data. All-subsets selection of enormous global models containing large numbers of predictors and their interactions makes analyses extremely prone to including uninformative parameters and 'overfitted' models. Burnham & Anderson (2002) caution strongly against all-subsets selection, and instead advocate 'hard thinking' about the hypotheses underlying the data. If adopting an all subsets approach, it is worth noting the number of models to consider increases exponentially with the number of predictors, where five predictors require $2^5$ (32) models to be fitted, whilst 10 predictors requires 1,024 models, both *without* including any interactions but including the null model.

Global models should not contain huge numbers of variables and interactions without prior thought about what the models represent for a study system. In cases where all-subsets selection from a global model is performed, it is important to view these model selection exercises as exploratory (Symonds & Moussalli, 2011), and hold some data back from these exploratory analyses to be used for cross-validation with the top model(s) (see Dochtermann & Jenkins, 2011 and references therein). Here, 90% of the data can be used to fit the model(s), with the remaining 10% used for confirmatory analysis to quantify how well the model(s) perform for prediction (Zuur & Ieno, 2016). Such an approach requires a huge amount of data (Dochtermann & Jenkins, 2011), but cross-validation to validate a model's predictive ability is rare and should result in more robust inference (see also Fieberg & Johnson, 2015).

Therefore, best practice is to consider only a handful of hypotheses and then build a single statistical model to reflect each hypothesis. This makes inference easier because the resulting top model set will likely contain fewer parameters, and certainly fewer uninformative parameters (Burnham & Anderson, 2002; Arnold, 2010). However, we argue all subsets selection may be sensible in a limited number of circumstances when testing causal relationships between explanatory variables and the response variable. For example, if the most complex model contains two main effects and their interaction, performing all subsets selection on that model is identical to building the five competing models (including the null model) nested in the global model, all of which may be considered likely to be supported by the data. A small number of models built to reflect well-reasoned hypotheses are only valid if the predictors therein are not collinear (see Collinearity section above). All-subsets selection using the R package *MuMIn* (*Bartoń,*

*2016*) will not automatically check for collinearity, and so the onus falls on the researcher to be thorough in checking for such problems.

## Deciding which information criterion to use

Several information criteria are available to rank competing models, but their calculations differ subtly. Commonly applied criteria include AIC, the small sample size correction of AIC for when $n/k < 40$ (AICc), and the Bayesian Information Criterion (BIC). Quasi-AIC (QAIC) is an adjustment to AIC that accounts for overdispersion, and should be used when overdispersion has been identified in a model (see Overdispersion section above). Note QAIC is not required if the overdispersion in the dataset has been modelled using zero-inflated models, OLREs, or compound probability distributions. *Bolker et al. (2009)* and *Grueber et al. (2011)* provide details of how to calculate these criteria.

Akaike's Information Criterion maximises the fit/complexity trade-off of a model by balancing the model fit with the number of estimated parameters. AICc and BIC both penalise the IC score based on total sample size $n$, but the degree of penalty for AICc is less severe than BIC for moderate sample sizes, and more severe for very low sample size (*Brewer, Butler & Cooksley, 2016*). Whilst AIC tend to select overly complex models, *Burnham & Anderson (2002)* criticised BIC for selecting overly simplistic models (underfitting). BIC is also criticised because it operates on the assumption that the true model is in the model set under consideration, whereas in ecological studies this is unlikely to be true (*Burnham & Anderson, 2002, 2004*). Issues exist with both AIC and BIC in a GLMM context for estimating the number of parameters for a random effect (*Bolker et al., 2009*; *Grueber et al., 2011*), and although degrees of freedom corrections to remedy this problem exist it is not always clear what method is being employed by software packages (see *Bolker et al., 2009* Box 3). *Brewer, Butler & Cooksley (2016)* show how the optimality of AIC, AICc and BIC for prediction changes with both sample size and effect size of predictors (see also *Burnham & Anderson, 2004*). Therefore, the choice between the two metrics is not straightforward, and may depend on the goal of the study i.e. model selection vs prediction, see *Grueber et al. (2011)* Box 1.

## Choice of ∆AIC threshold

Once all models have been ranked by an information criterion, it is common practice to identify a 'top model set' containing all models assumed to have comparable support in the data, normally based on the change in AIC values relative to the best AIC model (∆AIC). Historically, *Burnham & Anderson (2002)* recommended that only models with ∆AIC between 0 and 2 should be used for inference, but subsequent work has shown that for some models a much higher ∆AIC cut off is required to give a 95% probability of including the best (expected) Kullback–Leibler Distance model in the top model set (*Richards, 2008*; see also *Burnham, Anderson & Huyvaert, 2011*). An alternative approach to using ∆AIC cut offs is to include all models within a cumulative Akaike weight of $\geq 0.95$ from the top model in the '95% confidence set' (*Burnham & Anderson, 2002*; *Symonds & Moussalli, 2011*). Using high cut-offs is not encouraged, to

avoid overly complex model sets containing uninformative predictors (*Richards, 2008*; *Grueber et al., 2011*) but deciding on how many is too many remains a contentious issue (*Grueber et al., 2011*). We suggest Δ6 as a minimum following *Richards (2005, 2008)*.

## Using the nesting rule to improve inference from the top model set

It is well known that AIC tends towards overly complex models ('overfitting', *Burnham & Anderson, 2002*). As AIC only adds a two point penalty to a model for inclusion of a new term, *Arnold (2010)* demonstrated that adding a nuisance (completely random) predictor to a well-fitting model leads to a ΔAIC value of the new model of ~2, therefore appearing to warrant inclusion in the top model set (see section above). Therefore, inference can be greatly improved by eliminating models from the top model set that are more complex versions of nested models with better AIC support, known as the nesting rule (*Richards, 2005, 2008*; *Richards, Whittingham & Stephens, 2011*). Doing so greatly reduces the number of models to be used for inference, and improves parameter accuracy (*Arnold, 2010*; *Richards, 2008*). *Symonds & Moussalli (2011)* caution that its applicability has not yet been widely assessed over a range of circumstances, but the theory behind its application is sound and intuitive (*Arnold, 2010*).

## Using akaike weights to quantify variable importance

With a top model set in hand, it is common practice to use the summed Akaike weights of every model in that set in which a predictor of interest occurs as a measure of 'variable importance' (*Grueber et al., 2011*). Recent work has demonstrated that this approach is flawed because Akaike weights are interpreted as relative model probabilities, and give no information about the importance of individual predictors in a model (*Cade, 2015*), and fail to distinguish between variables with weak or strong effects (*Galipaud et al., 2014*; *Galipaud, Gillingham & Dechaume-Moncharmont, 2017*). The sum of Akaike weights as a measure of variable importance may at best be a measure of how likely a variable would be included in the top model set after repeated sampling of the data (*Burnham & Anderson, 2002*; *Cade, 2015*, but see *Galipaud, Gillingham & Dechaume-Moncharmont, 2017*). A better measure of variable importance would be to compare standardised effect sizes (*Schielzeth, 2010*; *Cade, 2015*). However, summed Akaike weights for variables in top model sets still represent useful quantitative evidence (*Giam & Olden, 2016*); they should be reported in model summary tables, and ideally interpreted in tandem with model averaged effect sizes for individual parameters.

## Model averaging when predictors are collinear

The aim of model averaging is to incorporate the uncertainty in the size and presence of effects among a set of candidate models with similar support in the data. Model averaging using Akaike weights proceeds on the assumption that predictors are on common scales across models and are therefore comparable. Unfortunately, the nature of multiple regression means that the scale and sign of coefficients will change across models depending on the presence or absence of other variables in a focal model (*Cade, 2015*).

The issue of predictor scaling changing across models is particularly exacerbated when predictors are collinear, even when VIF values are low (*Burnham & Anderson, 2002*; *Lukacs, Burnham & Anderson, 2010*; *Cade, 2015*). *Cade (2015)* recommends standardising model parameters based on partial standard deviations to ensure predictors are on common scales across models prior to model averaging (details in *Cade, 2015*). We stress again the need to assess multicollinearity among predictors in multiple regression modelling before fitting models (*Zuur & Ieno, 2016*) and before model-averaging coefficients from those models (*Lukacs, Burnham & Anderson, 2010*; *Cade, 2015*).

## CONCLUSION

We hope this article will act as both a guide, and as a gateway to further reading, for both new researchers and those wishing to update their portfolio of analytic techniques. Here we distil our message into a bulleted list.

1. Modern mixed effect models offer an unprecedented opportunity to explore complex biological problems by explicitly modelling non-Normal data structures and/or non-independence among observational units. However, the LMM and GLMM toolset should be used with caution.

2. Rigorous testing of both model fit ($R^2$) and model adequacy (violation of assumptions like homogeneity of variance) must be carried out. We must recognise that satisfactory fit does not guarantee we have not violated the assumptions of LMM, and vice versa. Interpret measures of $R^2$ for (G)LMMs with hierarchical errors cautiously, especially when OLRE are used.

3. Collinearity among predictors is difficult to deal with and can severely impair model accuracy. Be especially vigilant if data are from field surveys rather than controlled experiments, as collinearity is likely to be present.

4. When including a large number of predictors is necessary, backwards selection and NHST should be avoided, and ranking via AIC of all competing models is preferred. A critical question that remains to be addressed is whether model selection based on IT is superior to NHST even in cases of balanced experimental designs with few predictors.

5. Data simulation is a powerful but underused tool. If the analyst harbours any uncertainty regarding the fit or adequacy of the model structure, then the analysis of data simulated to recreate the perceived structure of the favoured model can provide reassurance, or justify doubt.

6. Wherever possible, provide diagnostic assessment of model adequacy, and metrics of model fit, even if in the supplemental information.

## ACKNOWLEDGEMENTS

This paper is the result of a University of Exeter workshop on best practice for the application of mixed effects models and model selection in ecological studies.

### Funding

Xavier A. Harrison was funded by an Institute of Zoology Research Fellowship. David Fisher was funded by NERC studentship NE/H02249X/1. Lynda Donaldson was funded by NERC studentship NE/L501669/1. Beth S. Robinson was funded by the University of Exeter and the Animal and Plant Health Agency as part of 'Wildlife Research Co-Operative'. Maria Correa-Cano was funded by CONACYT (The Mexican National Council for Science and Technology) and SEP (The Mexican Ministry of Education). Cecily Goodwin was funded by the Forestry Commission and NERC studentship NE/L501669/1. The funders had no role in study design, data collection and analysis, decision to publish, or preparation of the manuscript.

### Grant Disclosures

The following grant information was disclosed by the authors:
Institute of Zoology Research Fellowship.
NERC studentship: NE/H02249X/1.
NERC studentship: NE/L501669/1.
University of Exeter and the Animal and Plant Health as part of 'Wildlife Research Co-Operative'.
CONACYT (The Mexican National Council for Science and Technology).
SEP (The Mexican Ministry of Education).
Forestry Commission.
NERC studentship: NE/L501669/1.

### Competing Interests

Xavier A. Harrison is an Academic Editor for PeerJ. Beth S. Robinson is an employee of WildTeam Conservation, Surfside, Cornwall. The authors declare no further competing interests.

### Author Contributions

- Xavier A. Harrison conceived and designed the experiments, analysed the data, prepared figures and/or tables, authored or reviewed drafts of the paper, approved the final draft.
- Lynda Donaldson conceived and designed the experiments, authored or reviewed drafts of the paper, approved the final draft.
- Maria Eugenia Correa-Cano conceived and designed the experiments, authored or reviewed drafts of the paper, approved the final draft.
- Julian Evans conceived and designed the experiments, analysed the data, authored or reviewed drafts of the paper, approved the final draft.
- David N. Fisher conceived and designed the experiments, authored or reviewed drafts of the paper, approved the final draft.
- Cecily E.D. Goodwin conceived and designed the experiments, authored or reviewed drafts of the paper, approved the final draft.

**PeerJ** ___________________________________

- Beth S. Robinson conceived and designed the experiments, authored or reviewed drafts of the paper, approved the final draft.
- David J. Hodgson conceived and designed the experiments, authored or reviewed drafts of the paper, approved the final draft.
- Richard Inger conceived and designed the experiments, authored or reviewed drafts of the paper, approved the final draft.

### Data Availability

Figshare: DOI 10.6084/m9.figshare.5173396.

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
