# Peer review of "A brief introduction to mixed effects modelling and multi-model inference in ecology"

_PeerJ, doi:10.7717/peerj.4794_

## Round 0.1 · original submission · Major Revisions

I congratulate the authors for their ambitious undertaking in writing such a wide-ranging manuscript and for having achieved a number of successes in this task, a number which I think addressing the three reviewers' comments will only increase. All three reviewers have made supportive comments about your manuscript and indicated that this manuscript could make an important contribution to the literature, in particular to assist non-specialist practitioners, who surely must struggle to keep up with the constant flow of literature in this area even more than I do as a statistician.

In particular, I recommend that the authors carefully consider the suggestion of the third reviewer that the manuscript could be split into two, which would allow more detail to be provided in some areas (as suggested by the second and third reviewers) and also help with the second reviewer's comments about cohesion. However, the authors may also feel that they can respond to all reviewers' comments within a single manuscript.

One suggestion that I would like to add to the reviewers', at the risk of adding even more variety to the material already covered, is to consider a brief mention of software appropriate for such modelling. While R is well covered, I know people who work in this area using Stata and even SPSS, and I'm sure there are others who use SAS, etc.. If the authors do add a brief mention of other software options, I would also suggest that they consider providing a comment about the issue of different software sometimes producing different estimates due to different defaults and estimation approaches, especially for GLMMs, something that can be very disconcerting for non-statisticians and statisticians alike.

Reviewer 1 ·

Basic reporting

no comment

Experimental design

no comment

Validity of the findings

no comment

Additional comments

The review summarises the best practice and recommendations for ecologists conducting LMMs and GLMMs. Of course, we have a wonderful paper by Bolker et al (2009; Trends in Ecology and Evolution) but this was nearly 10 years ago and we probably need an update. I think this manuscript is a very approachable paper from which practicing ecologists can learn. I like seeing my good references written by ecologists which makes this paper really relevant to the target audience. I have several relevant references the authors.

Lines 104 and Line 346. The authors may have missed this paper:

Warton DI, Lyons M, Stoklosa J, Ives AR. Three points to consider when choosing a LM or GLM test for count data. Methods Ecol Evol. 2016;7(8):882-90.

Line 594. There is a recent paper

Nakagawa, S., Johnson, P. C. D. & Schielzeth, H. (in press) The coefficient of determination R2 and intra-class correlation coefficient from generalized linear mixed-effects models revisited and expanded. Journal of the Royal Society Interface (BioRxiv. doi: https://doi.org/10.1101/095851)

Line 640. There is a very useful simulator for ecologists and evolutionary biologists called SQuID

Allegue H, Araya-Ajoy YG, Dingemanse NJ, Dochtermann NA, Garamszegi LZ, Nakagawa S, et al. Statistical Quantification of Individual Differences (SQuID): an educational and statistical tool for understanding multilevel phenotypic data in linear mixed models. Methods Ecol Evol. 2017;8(2):257-67.

Lines 698 Murtaugh has several good papers on this topic. The latest I believe in this special issue.

http://esajournals.onlinelibrary.wiley.com/hub/issue/10.1002/ecy.2014.95.issue-3/

This relates to the next section of model selection as well.

Line 834

I wondered why the authors did not talk about DIC? (known to be not great for GLMMs although probably works for LMMs).

Line 887

Galipaud M, Gillingham MAF, Dechaume-Moncharmont F-X. A farewell to the sum of Akaike weights: The benefits of alternative metrics for variable importance estimations in model selection (online early). Methods Ecol Evol.

Reviewer 2 ·

Basic reporting

This paper provides an overview of methods and considerations in the use of mixed effects modelling in ecology. This is a timely subject, and likely to be of use to many readers. Overall I found that the paper contained many useful tips and suggestions. I have provided a number of comments that I believe could improve the utility and rigor of the paper.

One of my main concerns was a lack of cohesion amongst the topics. The paper seems as though the authors have attempted to cover too many issues, and as a result the most topics are covered only superficially (many of the topics are covered in greater detail in other reviews, which the authors cite), and the connections between topics are weak. Thinking about the title and objectives of the paper, I felt that the paper would be strengthened by a greater specific focus on considerations around random factors and apportioning and interpretation of error. For example, I thoroughly enjoyed reading the first half of the paper (up to L275) but felt more and more unsure of the goals of subsequent sections. For example,
• The section “choosing error structures and link functions” is too long, as much of this material is well covered by other reviews, and the considerations noted are not necessarily dependant on whether a research runs an LM or LMM. Focussing the current review on just a brief description of link functions, followed by a more detailed consideration of the impact of random factors on link function choice (if any), would be much more valuable.
• Similarly, the section “choosing predictors and interactions” misses an opportunity for a more in-depth discussion of decisions around which random factors to include – for example I found little guidance for researchers on how to make decisions about “nested” random factors, or how to decide whether it is appropriate to use multiple random factors additively (how is the error then distributed?). This section could be assimilated with the section “how complex should my global model be?”. Many of these issues are also relevant to the section “using the nesting rule to improve inference from the top model set” – how should researchers consider their use of random factors when thinking about model nesting?
• The section “assessing predictor collinearity” makes no mention of random factors at all, which is surprising given their primary importance to the paper (in the title), and the fact that researchers may wonder what to do about collinear random factors.

Some additional general suggestions for improvements:
• Around L470 would be a good place to comment on the impact and management of missing data, with particular reference to missing data amongst fixed versus random factors.
• Section “standardising and centering predictors” – this approach can also be considered as a transformation of the predictor variables, and so might be best if shortened and assimilated with the material at L294-301.
• L852 “at least delta-6 is required to guarantee a 95% probability…”, actually the use of a 95% confidence set, as described at L855 is the best way to guarantee 95% probability – I would suggest that choosing a delta cut off versus using sum of Akaike weights represent two different approaches to determining the final set. What is meant by a “cumbersome” (L856) top set, and why is this a problem?
• L942 “reversible jump MCMC” here in the last line of the paper is the first mention of this technique! I suggest providing more detail elsewhere, if it is relevant, or removing from the conclusions.

Overall the paper is well written, however I found the tone to be a informal in places. Specifically, the paper is written in the second person in many places, which comes across as prescriptive rather than informative, and is mostly unnecessary for statements that are not addressed to an individual. For example,
• L356 “you can’t transform a 0” should be “zeroes cannot be log transformed”,
• L483 “where you subtract the mean…” should be “where the mean is subtracted”.

The literature is well referenced and relevant.

The figures are of high quality and appropriate.

Experimental design

I appreciated the use of simulations to test some of the ideas being put forward. The text indicates at L662 that the R-code used to generate the simulations is available as online supplementary material, however I was unable to find this file amongst the review documents and therefore unable to confirm that the code runs as expected.

A L869 the authors “…suggest delta-6 as a minimum following Richards…” – but the authors conduct simulations to determine the false positive rate of only delta-6, why not also investigate delta-2, which is also commonly used, and then use the empirical evidence to support their qualitative arguments?

Considering the simulations, I’m not convinced that it is appropriate to use information theory to rank and compare models, and then using null-hypothesis testing to consider the significance of predictors and draw conclusions about the likelihood of type I error. Unless I am misinterpreting the methods, this seems to be mixing statistical paradigms in a way that many users would find inappropriate. Better would be for the authors to follow the advice put forward in the paper and use measures of model quality such as R2 to determine whether delta-6 (or any other delta value, see my comments above) provides the “best” model fit of their data. I would particularly like to see the impact that random factors have on the outcomes, given the objectives of the paper. Looking at the simulation methods, only one type of random factor was used “the random effect was always a six level factor in continuous blocks of random length”. Why not simplify the fixed factors, and experiment with single or multiple random factors, nested or unnested random factors, different types of response variables (and thus different link functions)? Taking this approach would enable the simulation material to quantitatively support many of the recommendations made in the paper.

Validity of the findings

While the paper collates some good recommendations from the literature, I found that a few recommendations need more evidence to be convincing. Some of this may come from greater use of the simulation study (see above). For example,
• L421 “good rule of thumb” – by what measure is it “good”? Is this rule proposed here for the first time (on what evidence?) or by others (provide citation). Is it plausible that this may not be a good rule in certain circumstances?
• L432 “…ideally >100” – what is this figure based on?
• L463 “…potentially arbitrary choice…” – why is it arbitrary? What is the choice typically based on? What should the choice be based on?
• L510-512 “often, researchers…claim that the large difference in AIC between the top and null models is evidence of a good fit” – do the authors have evidence that this practice is common?
• L557 “…can ‘fix’ the overdispersion…” – how/why?

Reviewer 3 ·

Basic reporting

largely okay to good (language, structure, self-containment); some references are under-appreciated (see detailed comments to authors)

Experimental design

doesn't really apply. The simulations are well designed but their evaluation misses a few points (see detailed comments to the authors).

Validity of the findings

doesn't really apply. However, the findings of the simulations seem valid (but their evaluation should be changed; see my detailed comments to the authors).

Additional comments

General comments

This manuscript gives an overview abouut two concepts/approaches that rapidly gained popularity in applied statistics during the past decade, namely mixed models (GLMM) and multi-model inference (MMI). The authors give a broad overview and treat a couple of issues coming along with the use of GLMM and MMI. Overall, I feel very sympathetic about the authors' intention to formulate 'best practice' guidelines for users of these approaches since more guidance as to how to use GLMM and MMI appropriately in ecology, animal behaviour, etc. is definitely needed. I also feel that the authors managed to cover the more recent literature fairly well and to write in a style which will be accessible to most scientists, also those not very familiar with statistical jargon.

However, I also feel that the goal of the article is too ambitious to be met. First, GLMM and MMI have in common that they only fairly recently became more popular and that they are based on maximum likelihood fitting, but apart from that there is not really much of a relation between the two approaches (as can be also seen in the manuscript). Hence, I feel the manuscript could benefit (perhaps a lot) by splitting it into two. This would allow for each of the two topics to be treated more in-depth, which, at occasions, would be highly desirable. In fact, as off now, the manuscript treats several crucial issues quite briefly (too briefly, in my view; see below for details). Second, the title ('Best practice...') sets the level very high, but the manuscript then quite frequently remains fairly vague and does not go much beyond mentioning open questions (examples are, e.g., random slopes or the choice among AIC and BIC; see below). In parts, this is not surprising (and probably the most and best the authors can do) since GLMM as well as MMI are both methods which currently are subject to developments and improvement as well as investigations of their properties (with sometimes surprising outcomes). As a consequence, our understandings of how to use and interpret results of GLMM and MMI have changed quite drastically over the past years (e.g., importance random slopes in GLMM or use of Akaike weights for predictors in MMI; see below). I myself use both approaches very regularly and frequently and feel fairly well on top of the current literature but still would not dare to formulate 'best practices' for either of them, feeling that they might be outdated in the near future and/or some would be matter of too much uncertainty and debate. (so regarding the title I'd feel beginning it something like 'A brief introduction to …' would probably be better suited than 'Best practices...'.). For further more detailed comments see also further below.

Apart from that the authors at occasions lack clarity regarding the topics they treat. This is the case in the section about LRT, minimal adequate models and stepwise model selection (see my comments re L 675-703); in the section about all subsets and dredging (see my comments re L 760-817); in the section about the simulation addressing the effects of dredging (see my comments re L 772-788); when the authors refer to model selection (see my comments re L 809-811; to some extent this refers to the whole section, i.e., L 666-744). One might argue these issues to be mere semantics, but in my view most of them reflect fairly important sources of confusion which need to be worked out way better in order to not make things worse than they currently are (and this is particularly important for everything related to model selection).

Some more issues I feel being important can be found in the specific comments below.

Specific comments

general: I missed a statement about model stability

L 80-83: well described/presented; but the authors do not address how these issues can be tackled...

L 131-133: do they really model variance hierarchically in the sense of sequentially? I believe to recall from the literature that all these effects are estimated simultaneously (i.e., each while controlling for the others).

L 110-140: somewhere within this section it should be mentioned that the whole question of whether to fit a predictor as a fixed or random effect is only one for *factors* (i.e., quantitative predictors are always modelled as fixed effects).

L 146-186: this whole section falls short of what we know (and don't know) and need to do with regard to random slopes. First of all, in my view, whether random slopes are to be included should not so much be determined by the 'goals of the analysis' (L 152) or what we 'wish' to know (L 158-168), but should be driven by the data structure and the desire to have unbiased standard errors and confidence intervals for the fixed effects and keep type I error at the nominal level (of, e.g., 5%). In fact, there is now ample evidence that neglected random slopes can lead to drastically inflated type I error rates (see, besides Schielzeth & Forstmeier 2009, e.g., Barr et al. 2013. Random effects structure for confirmatory hypothesis testing: Keep it maximal. J Memory Lang, 68, 255-278; or Aarts et al. 2015. Multilevel analysis quantifies variation in the experimental effect while optimizing power and preventing false positives. BMC Neurosci, 16, 94). At the same time there are now articles emerging that recognize difficulties and potential disadvantages coming along with the use of such 'maximal models' (see, e.g., Matuschek et al. 2017. Balancing Type I error and power in linear mixed models. J Memory Lang, 94, 305-315; or https://arxiv.org/abs/1506.04967). So, rather than formulating 'best practice' (which, in my view, would be to generally consider random slopes), this section simply mentions the existence of random slopes and explains what they model. But it neglects that random slopes play a central role when wanting to infer about the relevance of fixed effects; and it misses the ongoing debate about them and the potential problems a maximal model can cause with regard to inference about random effects (and perhaps power for the fixed effects, too).

L 188, 206, 234: I found these headers kind of odd ('To …').

L 212: wouldn't one consider the intercept to be a fixed effect, too?

L 270-272: this second example I didn't understand.

L 232: I would not say that the transformation of the linear predictor 'normalises the residuals of the model', and I also don't really understand this statement.

L 353: I'd mention which assumptions need to be checked.

L 355-358: I was surprised to read this. What I see most frequently is that a constant (i.e., the same value) is added to *all* values.

L 408-411: it is not even possible to fit a model comprising 'more parameters than observations'. At the same time a model can also be overparameterized when it comprises fewer parameters than observations (see Box 8.3 in Quinn & Keough).

L 421-424: this rule of thumb I didn't understand at all. Failing in which sense?

L 436-437: I would be a little cautious with referring to the paper by Zuur et al. In the context here largely because it doesn't make at all clear that the practice they recommend for determining which interactions to include means to conduct an exploratory data analysis with all its consequences (elevated type I error rate, biased P-values, SE, and confidence intervals, etc.).

L 445: I'd say collinearity *among*.

L 457-462: In my few, VIF are to be preferred over correlations since heavy collinearity can come in forms undetectable when using correlations (that is, one should always inspect VIF). Also, there is a multitude of thresholds mentioned for VIF being critical (see, e.g., Quinn & Keough 2002 which mention a value of 10).

L 462-469: the solutions mentioned do not solve the problem of the arbitrary threshold.

L 468: a single or few.

L473-478: this is pretty imprecise. The 'outcome' mentioned is one of a PCA or FA but not of a model with two collinear predictors; and the mere fact that two variables 'share contribution to a principal component' cannot be 'taken as strong indication that the predictors’ signal cannot be teased apart'. In fact, it is a common misinterpretation that two variables having somewhat larger absolute loadings on a principal component must be strongly correlated.

L 443-478: this whole section does not mention at all how one would check for collinearity in a GLMM framework. This would be interesting to be treated since many users of GLMMs are concerned about collinearity between fixed and random effects, and because the R functions I am aware of cannot be applied to GLMMs.

L 489-490: when the main effects are covariates, or?

L 497-498: how could scaling improve the robustness of a model? The model is actually virtually identical (but the coefficients get another interpretation).

L 507-508: I don't see the link between model fit and robustness (a model can have a poor fit but be robust, e.g., in the sense of not suffering from instability; and a model can also show a very good fit but not be robust, for instance due to model instability or overfitting).

L 521: 'non-linear effects of the predictors on the response' to be precise.

L 526-528: but the inspection of residuals vs. fitted values only makes sense for Gaussian models, or?

L 534: 'means' rather than 'modes'?

L 536: random effect 'residuals' or rather 'means' ('BLUPs'; see Baayen RH. 2008. Analyzing Linguistic Data. Cambridge University Press. Cambridge).

L 551-553: overdispersion affects standard errors but *not* the estimates (Gelman A & Hill J. 2007. Data analysis using regression and multilevel/hierarchical models. Cambridge University Press. Cambridge; P 115).

L 657-663: this seems like an excessively complex way of examining absence of overdispersion. Why not taking the dispersion parameter of 3.16 as conclusive evidence of strong overdispersion?

L 695-699: Mundry (2011) is actually arguing against model simplification

L 691-694: I would say that the most important source of unreliable LRT of fixed effects in GLMMs are neglected random slopes (see my comments above).

L 675-703: this para mixes as couple of, at best, loosely connected topics. One topic treated is selection of a MAM, and, as shown by, e.g., Mundry (2011), this likely leads to greatly inflated type I error rates if the model or the terms it comprises are then tested using NHST. The, in my view, most important question is as to when to even consider an MAM? For testing it leads to greatly inflated type I error rates (Forstmeier & Schielzeth 2011; Mundry 2011) and I only see its value when it comes to making predictions. Another topic treated is stepwise model selection (as one method of coming up with an MAM; there are others), and this has so many times been shown to be useless that I don't see what else can be said about it rather than it being useless (see also Mundry & Nunn. 2009. Stepwise Model Fitting and Statistical Inference: Turning Noise into Signal Pollution. The American Naturalist, 173, 119-123). In between the authors briefly touch on the reliability of LRT for fixed effects in GLMMs (see my comment re L 691-694). And yet another topic is NHST itself (which could be contrasted with information theoretic or Bayesian inference). So I think this para is not helpful as it is. First, the authors need to decide which topic they want to address, and then they should focus on this topic.

L 719: I'd say 'similarly' rather than 'equal'.

L 749: I wouldn't label this approach 'conservative' but simply one which produces nominal type I error rates.

L 754-756: here the authors mix NHST and inference based on an information criterion which should never, never, never, be done. In fact, just browsing the book for statements written in bold, one can find several statements strongly arguing against using words such a 'significance' in the context of an IT analysis; and Mundry 2011 clearly showed that selecting a best model using AIC and then considering its significance or the significance of the predictors it comprises leads to greatly inflated type I error rates. Hence, such practice doesn't have any merits, and should clearly argued against.

L 760-817: this whole section is somewhat unclear in the sense that 'all subsets' and 'dredging' are conflated: 'all subsets' refers to how the set of models considered is constructed whereas 'dredging' refers to what is done with the results revealed for multiple models. The authors are correct that Burnham and Anderson (2002) repeatedly and strongly argued against 'all subsets' (but at the same time they report at least one such analysis in their seminal book); but others have been more relaxed about this (see, e.g., Box 2 in Stephens PA, Buskirk SW, del Rio CM. 2007. Inference in ecology
and evolution. Trends Ecol Evol, 22, 192-197). So in my view, no simple statement can be made regarding whether an 'all subsets' is advisable or not. 'dredging', in turn, refers to an analysis during which a couple of models (which may or may not have been 'all subsets' of a full model) have been fitted and inference is then drawn on some 'best' model (selected using stepwise or based on an information criterion) or simply the most (or only) significant one identified by a researcher. So these two topics are only loosely linked and the manuscript would benefit from the two treated separately.

L 772-788: this simulation conflates three issues: model selection and subsequent testing (as has been addressed by Mundry 2011); cryptic multiple testing of multiple predictors in a single model (as has been addressed by Forstmeier & Schielzeth 2011); and neglecting to test the contribution of a factor with several levels before considering pair-wise comparisons (see 'testing sets of variables' in Aiken & West. 1991. Multiple Regression: Testing and Interpreting Interactions. Sage. Newbury Park; or Cohen & Cohen. 1983. Applied Multiple Regression/Correlation Analysis for the Behavioral Sciences (2nd ed.). Lawrence Erlbaum Associates, Inc. New Jersey). All three issues lead to inflated type I error rates and the simulation doesn't tease these effects apart. So I think this simulation is not very helpful.

L 789: I think dredging should simply not be used.

L 801-802: one model to represent all hypotheses simultaneously, or one model for each of them separately? In the former case I would not see much of a need of applying any model selection at all...

L 804-809: here the authors should be careful: after having conducted a full-null model comparison sensu Forstmeier & Schielzeth (2011) this would no more represent dredging; but without a preceding full-null model comparison this would be dredging with all its adverse consequences (inflated type I error rate and effect sizes, etc.).

L 809-811: useful for which purpose? Here the authors stand on weak philosophical grounds. The first and most important question to be asked when it comes to model selection is 'what is it used for?' Model selection for inference and model selection for prediction are two very frequently confused but very different issues with very different implications and dangers. Here the authors are not clear about what they refer to and I fear they do not know either. So this needs to be reworked after it has been clarified what purpose (inference or prediction) the model selection serves.

L 848: why does model averaging 'require the identification of a “top model set”'? Its one of the beauties of this approach that one can simply average across all models (and models with little support then would have little impact on the model averaged parameters)... This has impact for the rest of the paragraph which actually seems rather addressing the question of how to draw inference in an MMI framework.

L 854-857: what is the problem with a 'cumbersome' top models set? It basically means that there is not a single of a few models that stick out from the rest – which, in my view, would be an important finding in itself.

L 874-877: I don't really see the problem here: the models are dropped from the set and then one recalculates Akaike weights. Or do I miss something here?

L 882-887: in this context two recent articles might be worth considered:
Galipaud, M., Gillingham, M.A.F., David, M. & Dechaume-Moncharmont, F.X. 2014. Ecologists overestimate the importance of predictor variables in model averaging: a plea for cautious interpretations. Methods in Ecology and Evolution, 5, 983-991.
Galipaud, M., Gillingham & Dechaume-Moncharmont, F.X. 2017. A farewell to the sum of Akaike weights: The benefits of alternative metrics for variable importance estimations in model selection. Methods in Ecology and Evolution, in press.
Both seem to have the potential to drastically change our assessment of the value of Akaike weights for inference about predictors.

L 925: fishing expeditions *are* very risky.

---

## Round 0.2 · Major Revisions

I would like to thank the reviewers for yet again providing extremely thoughtful and useful comments on this manuscript. I feel that the authors have set themselves a very ambitious task in covering so much material in a single manuscript, even with possible future manuscripts to flesh out aspects that cannot be fully addressed here, in a technically correct manner whilst preserving the necessarily readability for their audience, and responding to the reviewers’ comments will help in achieving this goal.

While your preference for a single manuscript is clear, you may of course wish to revisit the option to split this manuscript into two separate manuscripts in your response at this stage. However, I also appreciate that we risk running in circles over the same ground with this, and perhaps some other topics, and so I will ask that you carefully consider all of the reviewers’ points and for each, either make changes or provide a justification for not making changes. I agree with the reviewers that the revised version of the manuscript is stronger and hope that it will become stronger still with further revision.

I will draw the authors’ attention in particular to the (major) points 2, 3, and 4 from the third reviewer, which need careful consideration, alongside carefully addressing reviewer two’s first point. Reviewer two’s third point raised an important practical issue (ID labelling) that I think is well worth addressing as I regularly encounter PhD candidates who are unclear whether they are numbering within or across clusters. Needless to say, all of the three reviewers have provided valuable feedback and all of their comments should be addressed.

I think that, in general, you have done well to avoid many overly definite statements in the manuscript, but at line 121, I wonder if a caveat along the lines that “As with all heuristics, there will be situations where these recommendations will not be optimal. If the researcher has concerns about the appropriateness of a particular strategy for a given situation, they should consult with a statistician who has experience in this area.” would be useful to remind the reader that relatively few rules, if indeed any, in statistics are universal and to encourage them to collaborate with specialists as appropriate. This would not, of course, relax requirements around technical correctness, but it would, I hope, alert the reader to proceed with due care in this challenging area.

Reviewer 1 ·

Basic reporting

no comments

Experimental design

no comments

Validity of the findings

no comments

Additional comments

I have reviewed an earlier version of this MS. The authors have addressed my comments and it seems they did so for others' comments. I have noticed that one of the other referees raised a point about missing data, which I did not comment on. Actually, missing data on mixed models are still underdeveloped and an emerging research topic. There are a couple of papers that directly discuss missing data in mixed model contexts, but which are not currently cited in the MS but relevant.

Nakagawa, S. (2015) Missing data: mechanisms, methods and messages In: Ecological Statistics: contemporary theory and application (eds. Fox, G. A., Negrete-Yankelevich, S. & Sosa, V. J.). Oxford University Press, Oxford. pp. 81-105

Noble, D. W. A. & Nakagawa, S. (submitted) Planned missing data design: stronger inferences increased research efficiency and improved animal welfare in ecology and evolution. bioRxiv

https://www.biorxiv.org/content/early/2018/01/11/247064

The latter one includes an interesting example of how planned missing data design can be used in a repeated measurement design (which is modelled by mixed models). I thought the authors may be interested. No need to cite these but the authors would like to note that missing data on mixed models are topics under development.

Reviewer 2 ·

Basic reporting

Language and presentation are appropriate

Experimental design

Not applicable

Validity of the findings

Not applicable

Additional comments

In this paper, which I reviewed previously, Harrison et al. provide an overview of considerations associated with (generalised) linear (mixed) modelling in ecological datasets. Overall the paper has much improved from the previous version, and will likely make a valuable contribution to the literature. I feel that a few of my previous comments still apply to the new version, so I hope the details below clarify my concerns for the authors:

1. Regarding the simulation analysis, I accept the authors’ response about choosing not to run more models. However, I am still concerned about the mixing of NHST and IT philosophies in the interpretation. Specifically, there is heavy reliance on interpreting results as “Type I error” – strictly speaking one cannot (or perhaps, should not) falsely reject a true null hypothesis with an information theory approach, because one is not doing null hypothesis significance testing. Of course, I agree with the authors that it is important to consider how often one might draw a false conclusion about whether a given variable is an important predictor of a response. However, it is important not to assess models built under IT using NHST. A better approach would be for the authors to use the inference strategies they advocate elsewhere in the paper, such as testing model assumptions, reporting R2, and basing interpretation on the presence and size of effects in the final model (see also my comment below on RI) – would these metrics collectively lead a researcher to erroneously conclude that they had settled on an appropriately parameterised final model that suggests strong effects? Similarly, the sentence at L944-945 presents a very frequentist viewpoint on model selection, which is inappropriate in a discussion of information theoretic approaches to model selection.

2. Regarding the use of delta-6 or 95% summed weights, I felt that the authors response to my previous comment missed my point somewhat, which is that delta values and summed weights represent two (albeit slightly) different approaches. If one wants to “guarantee” 95% weight, then they should use 95% summed weight. For example, in the paper cited by the authors, model “M4” required delta-6.5 to encompass the best EKLD model 95% of the time (Richards, 2008, p223). While “at least delta-6” may be a good general guideline, this is not the same as “guaranteeing 95% weight”. See also Burnham and Anderson, 2002, p78, for explanation of how delta values represent relative evidence.

3. New material has been added on the specification of random factors: I believe the lme4 specification “Clutch Mass ~ Foraging Rate + (1|Woodland) + (1|Female ID)” should achieve the same goal as the model on L402 (should this be M5?), if all female IDs are unique. Ambiguity can arise if the females are labelled (say) 1 – 10 in woodland 1, and 1 – 10 in woodland 2: are there 10 females, each breeding at two sites, or 20 females? Labelling females as 1 – 20 (if the latter) alleviates the ambiguity between crossed and nested random factors, and it would be worthwhile explaining this. See for example lme4.r-forge.r-project.org/book/Ch2.pdf

4. Considering the revised section 5 (p34), I think the authors are overly dismissive of “relative importance” statistics. Unfortunately, the vernacular definition of “importance” differs somewhat from the technical definition; “relative importance” (sum of weights) informs which parameters are likely to occur in the best model, not their effect sizes. Indeed, IT-based model selection determines two things: the probability that variables should be present in the model (taking model selection uncertainty into account) and, if variables are included, what their effect sizes are. RI can assist inference around the former. I accept that there is discussion about the effectiveness and interpretation of RI in the literature (e.g. Giam and Olden, 2016; Galipaud et al., 2017), but I am concerned that the tone of section 5 might suggest to readers that they should ignore RI altogether. My view is that inference in an information theoretic framework should take into account (or at least, report!) all measures of quantitative evidence about the models, including RI and parameter estimates. For example, a predictor with RI of 1 and a small effect size may or may not be biologically “important”, while a low RI would likely indicate poor support, even if an associated effect is large. These statistics are additional tools for inference.

5. A note on the suggested model specifications (p5, p6, p8): it is great these are provided, as knowing how to specify a model in R can be very useful to new users. These models are set within a discussion that considers the differences among models, so it would be helpful to see how the model outputs change depending on the model specifications. This could be easily obtained using a “built in” dataset in R, and presented as supplementary material with a brief commentary on how to interpret each output. For example, the output from M1 and M2 could be narrated in terms of the interpretation of output values associated with “group”. The same approach could be applied to M3 and M4. As an aside, a more complete discussion of for the use of random slopes would be provided if the authors explain the circumstances under which one would prefer model M4 over a model specified as “glmer(successful.breed ~ 1 + (body.mass|sample.site)”, which is sufficiently similar to model M2 that I think it is worth noting (see also e.g. Gelman and Hill, 2007, p259).

6. A note on whether removing models from the set makes interpretation of the Akaike weights “difficult” (L1011): as pointed out by Burnham and Anderson (2002, p75), adding and removing models from a set requires weights be recalculated. From a practical perspective, it is straightforward to calculate the weights of any model set by manually making the list of models for further analysis with MuMIn, or by using ‘get.models’ (MuMIn) on an existing ‘dredge’ object – see the MuMIn documentation for more info. In fact, wi values can even be manually recalculated relatively easily too (using the formula in Burnham and Anderson, 2002, p75). Explicitly specifying the model set is what the authors are indirectly recommending when they suggest that one should consider whether the “all-subsets selection” approach is appropriate, and so mentioning these methods for re-calculating weights would overcome the “difficulty” and provide readers the flexibility of choosing their own model set.

Minor comments

L105 “best practice”, in light of the title change, best to change this phrase here too?
L369 “checking the assumptions of the LMM or GLMM is an essential step” – given the high level of detail and instruction provided elsewhere in the paper, I felt like this statement needs a sentence or two also providing suggestions for some of the methods/packages that could be used.
L641 note that it is also possible for data to be under-dispersed (less variance than expected by chance, i.e. more uniform than chance).
L719 isn’t a Gaussian GLMM a LMM?
L769 a dataset cannot be “made up of missing data” – suggest rephrasing
L839-840 “there is no ‘null’” – I see the authors’ point, although one occasionally sees the intercept-only model referred to as the ‘null model’ – suggest rephrasing.
L1062 “data dredging” and “fishing” are no longer major components of the paper, and could therefore be removed from the Conclusions?
The paper contains a lot of abbreviations – some of which are widely used, and others less so. I suggest the authors consider providing a glossary of abbreviations. Furthermore, many abbreviations that are used only once or twice are probably unnecessary, and the paper would be more readable if the terms were spelt out e.g., “GLS” (L633), “IT-AIC” (L775) “ASS” (L897 and 898).


Literature cited in this review

Burnham, K.P., Anderson, D.R. (2002) Model selection and multimodel inference: a practical information-theoretic approach. Berlin, Springer.
Galipaud, M., Gillingham, M.A.F., Dechaume-Moncharmont, F.-X. (2017) A farewell to the sum of Akaike weights: The benefits of alternative metrics for variable importance estimations in model selection. Methods in Ecology and Evolution, 8, 1668-1678.
Gelman, A., Hill, J. (2007) Data analysis using regression and multileval/hierarchical models. Cambridge, Cambridge University Press.
Giam, X., Olden, J.D. (2016) Quantifying variable importance in a multimodel inference framework. Methods in Ecology and Evolution, 7, 388-397.
Richards, S.A. (2008) Dealing with overdispersed count data in applied ecology. Journal of Applied Ecology, 45, 218-227.

Reviewer 3 ·

Basic reporting

all okay to good.

Experimental design

doesn't really apply with the exception of the simulation which is okay, but doesn't really much and conflates mode selection with testing (see my comments to the authors)

Validity of the findings

doesn't apply

Additional comments

General comments

This manuscript greatly improved as compared to the last version, and I appreciate the authors' efforts to deal with my comments and recommendations. However, I still see need for improvement and refinement. More specifically I see four major points (in my view issue two and four are the most crucial ones):

* First, I still feel that the manuscript's two major topics, GLMM and MMI, are not really connected, neither regarding statistical theory nor in the manuscript itself, and that the manuscript would benefit from being split into two. Moreover, I even see dangers in presenting both approaches in the same manuscript. In fact, as the authors themselves (correctly) state, the number of estimated parameters (degrees of freedom) in GLMM are generally unknown (L 883-884). This, in turn poses the question of how to even determine an information criterion such as AIC or BIC which penalizes for model complexity. The authors do not elaborate on this question, beyond briefly mentioning the issue. In their cover letter the authors state that they want to present the entire 'analysis pipeline'; but to me it seems that this particular pipeline (i.e., GLMM in combination with MMI) does not rest on a solid theoretical basis (and the fact that the two methods are frequently combined in applied statistics in ecology and behaviour doesn't alleviate the problem). So maybe better not presenting as being logically following one another, perpetuating common practice but not promoting best practice.

* Second, I still see need for refinement regarding the sections about random slopes (particularly L 379-402). Although the authors have considerably improved the manuscript regarding this point, the current manuscript still cuts the issue too short. The key point is that the authors seem to mix random slopes and correlations among random intercepts and slopes. In fact, the maximal model proposed by Barr et al. (2013. J Memory Lang, 68, 255–278) represents a model comprising random intercepts, all possible random slopes and also all correlations among them. The authors of the current manuscript are correct in stating that (nearly) perfect correlations are indicative of a model being too complex for the data at hand. However, the situation is not an all-or-nothing. In fact, when the correlations cannot be reasonably estimated, it might still be possible to fit a model including random intercepts and all random slopes but not the correlations among them (see Bates et al. 2015. Journal of Statistical Software, 67, 1-48). This is also the approach somewhat proposed by Barr et al. (2013) and Bates et al. (Parsimonious Mixed Models. http://arxiv.org/abs/1506.04967v1): begin with the most complex model but simplify it, beginning with the exclusion of correlations among random intercepts and slopes when it appears too complex.
There is one additional point I feel the authors must make: it should not be a matter of taste whether random slopes are considered. Instead, based on the available literature Schielzeth & Forstmeier (2009. Behav Ecol, 20, 416-420), Barr et al. (2013), and Aarts et al. (2015. BMC Neurosci, 16, 94) it simply seems needed to account for random slopes to achieve the nominal type I error rate (and correspondingly unbiased standard errors and confidence intervals); and when all or several random slopes are neglected because the model would get too complex to fit it, then one has to face a perhaps highly elevated type I error risk.
Also at other occasions (e.g., L 172-187), the authors still leave the impression that decisions regarding whether to include random slopes are a matter of 'taste' or the aims of the study, but in my view Schielzeth & Forstmeier (2009), Barr et al. (2013), and Aarts et al. (2015) made convincing cases that these have to be included to prevent type I errors.

* Third, I feel concern about the section headed 'Stepwise Selection, Likelihood Ratio Tests and P values' (716-750). In my few this conflates a couple of issues in an inappropriate and confusing way. First, NHST is one of the three major statistical 'philosophies' (together with Bayesian and information theory based inference) used to draw inference about a model and/or individual predictors. Stepwise procedures, in turn, are a specialized technique which had been (and unfortunately still is) used as a means of model simplification; stepwise can be based on NHST, but also on an information criterion like AIC or other criteria (e.g., an F value). As such NHST has nothing to do with stepwise procedures, and the far, far majority of uses of NHST have nothing to do with stepwise selection. As the authors correctly state, both have come under heavy criticism, but for very different reasons. NHST is criticised since decades for the many weaknesses and pitfalls the approach has; stepwise has been criticised for a couple of particular weaknesses (e.g., Whittingham et al. 2006. Journal of Animal Ecology 75, 1182–1189) among which is an extremely inflated type I error probability (Mundry & Nunn. 2009. Am. Nat., 173, 120-123).
Here are a couple of clarifications which seem needed:
- NHST, when not coupled with stepwise and used appropriately in the sense of Forstmeier & Schielzeth (2011. Behav. Ecol. Sociobiol., 65, 47–55) does not lead to overestimated effect sizes or an inflated type I error rate.
- the null model sensu Forstmeier & Schielzeth (2011) is not one that differs from the full model by a single predictor, but by a set of predictors and the comparison between the two reveals a global test of their combined effect (appropriately accounting for multiple testing which otherwise would be an issue in case one would base inference on the significance of the individual terms in the model without conducting a full-null model comparison).
- Mundry (2011) did not argue in favour or model simplification.
- Murthaugh (2009) found that stepwise and all subsets approaches revealed largely comparable results (wrt their predictive performance and number variables selected) and regardless of whether stepwise was based on an F-test or information criteria. As such the article does not really compare NHST with other philosophies but compares model selection techniques.

* My last major concern is about the fact that the authors lack clarity and rigour regarding the issue of mixing model selection with significance testing. In fact, Burnham & Anderson (2002) have repeatedly pointed out that mixing the two philosophies is a no-go (and Mundry 2011 showed that it leads to drastically inflated type I error rates). For instance, when applied appropriately, there is no such thing as a type I error when using model selection based on an information criterion (such as AIC) because in the context of such an analysis one simply must not use any tests (see Burnham & Anderson (2002), e.g., P 202-203). In the view of the authors of the current manuscript, however, such an option seems to exist (see, e.g., L 805-807; 850-852; 855-859; caption of Fig. 4). Its worth emphasizing here that the term 'test', as I used it here, also encompasses checks of whether confidence intervals comprise the zero (and that bias in P-values gets along with parallel biases in standard errors (being too small) and effect sizes (being too large)).

Apart from that, I am still not very convinced by the simulation (L 813-836), since it conflates a couple of issues, namely (i) combining model selection with significance testing (see also my previous comment), (ii) conducting pair-wise comparisons without a global test of the effect of a factor, and (iii) neglecting the full-null model model comparison). Each of these have been addressed more thoroughly and clearly in other papers (i: Burnham & Anderson 2002; Mundry 2011; ii: all stats books covering ANOVA; iii: Forstmeier & Schielzeth 2011), and I feel this section does not contribute much to the existing literature nor to the clarity of the manuscript. On the other hand, the authors should make clearer statements about not combining model selection and significance testing (in a wider sense, i.e., encompassing also inspection of whether confidence intervals encompass zero; see also my previous comment).

Specific comments

L 33: Pretty abrupt transition from one topic to another.

L 67-68: inference 'about'.

L 69-72: I felt these examples are confusing: if the fixed effect varies or is manipulated at the clutch level, how can pseudo-replication happen on the chick level (it happens on the clutch level if each comprises several chicks, each measured once). And 'if fixed effects vary at the level of the chick': between or within chick? In case of the former the authors are correct that non-independence occurs among clutches or mothers'; but in case of the latter it also happens on the chick level.

L 145: 'hierarchically' implies that certain effects get priority when being estimated; but to my knowledge all effects are modelled/estimated simultaneously.

L 165-167: in my view random slopes need to be included to keep type I error rate at the nominal level of 0.05 and obtain unbiased standard errors and confidence intervals.

L 179-182: it seems worth mentioning that this model comprises the random intercept, the random slope and also the correlation between the two.

L 183: Schielzeth & Forstmeier (2009) *show* that...

L 274-279: I don't get the point here. Certainly, the random effects structure needs to be set up appropriately; but as Barr et al. (2013. J Memory Lang, 68, 255–278) showed, P-values for individual effects should best be based on likelihood ratio tests comparing the full with respective reduced models. This, in turn, doesn't rely on determination of residual degrees of freedom.

L 285-287: *general* linear models make the assumptions of normality and homogeneity of residuals (in my view, 'linear models' is a generic term encompassing also Generalized Linear Models such a Poisson or logistic models).

L 287: lower case 'normality'.

L 308-318: my intuition tells me that those not being already familiar with the link function wouldn't get this section.

L 372-376: but this applies in general (not only to GLMM).

L 374: 'Crossed factors *allow to* accurately estimate...'.

L 566: Zuur et al *give*.

L 586: 'estimates *of* …'.

L 658-659: to my knowledge, Wald-/t-tests are not applicable for random effects (and maybe not even F-tests).

L 694: 'often' or 'always'.

L 779: 'reference*s*'.

L 788: '...details on *how* AIC...'.

L 792-795: comparing the full with a null model is not an alternative to NHST, but simply NHST applied appropriately (avoiding type I errors due to multiple testing)..

L 805-807: but this applies only when one selects the best model and then tests it, something which should never be done as repeatedly and strongly stated by Burnham and Anderson (2002).

L 817: 'ASS' not formally introduced.

L 818: I guess you the authors mean 'different' rather than 'separate'.

L 817-819: I don't see any difference: what the function dredge of the package MuMIn does is exactly dredging in the sense of fitting all possible subsets of models...

L 849-852: as Burnham and Anderson (2002) have pointed out repeatedly one should not use the term 'significant' in the context of an AIC-based analysis (e.g., P 84, 203), and I feel the authors should adhere to this rule.

L 855-860: another example of the authors mixing model selection and significance testing. According to Burnham and Anderson (2002) such exercise must not be done (see previous comment), and Mundry (2011) clearly showed how such practice leads to drastically inflated type I error rate.

L 859: which 'conditions'?

L 930: 'included' in what? The AIC-best model?

L 936: I'd use 'similar' rather than 'equal'.

Figure 1, caption: reference to (B) is missing. The fact that the overall intercept is 0 is irrelevant here.

Figure 2, caption: I'm not sure if I would speak of a biased estimate when its average seems pretty much perfectly matching the simulated value (I'd speak of 'bias' then the average deviates from the simulated value).

---

## Round 0.3 · Minor Revisions

I would again like to thank the reviewers for their thoughtful re-reading of this interesting and useful manuscript. Reviewer 3 has raised a number of queries, some of which I fear are a result of the broad coverage within a single manuscript and others which I think will assist you in ensuring careful and precise language throughout.

When reading through the manuscript again, I was frequently tempted to suggest extensions or clarifications, but these would easily double, and perhaps triple, the size of the manuscript. One possibility to respond to reviewer 3's two main points would be to add a few sentences to each section, not to do justice to the points they are raising, but at least to make the reader aware that this "sampler" manuscript by necessity has to focus on a (relatively) small number of issues, albeit most of the main ones I think, and that there is much, much more to consider.

A few additional comments from me for you to also consider are:

Line 504: I wonder if it would be worth adding "and its daughter terms" after "a 3-way interaction between continuous predictors". While the inclusion of all 2-way and main effects in this context is almost always required, there are exceptions which would result in fewer than 8 parameters. I'd also add "using this rule" to the end of this sentence, as you've noted other heuristics immediately before.

Line 561: I'd suggest "*more easily* interpretable" as the main effects are always interpretable as long as one remembers exactly what they represent.

Line 567: I'd suggest "indeed *sometimes* recommended" as this otherwise sounds like an absolute recommendation. For variables with similar scales already, scaling is unlikely to offer any benefits whatsoever.

Line 637: If you don't mind a stylistic suggestion, I'd change the second "but" to an "and" to avoid the repetition.

Line 660: I'd add "multiple" before "so-called" to emphasise the point, and I think this is one issue that novice modellers quickly run into with pseudo R-squared statistics, that this is no single natural measure here.

Line 698: I think I'd add a parenthetical reference to "residual" as another version of the R in REML. For completeness, you could also add "reduced" but I seldom see this alongside the roughly 50-50 mix of restricted and residual forms.

Line 715: I tried to avoid suggesting additions, but a brief mention of Kenward and Roger's approach here would be useful (I like it's properties in small samples).

Line 753: There is much you could add around missing data, but I think it is important to encourage the reader to at least think about how missing data is the (mostly stochastic) result of missing data mechanisms. I'd add "depending on the missing data mechanism" before the "and reduces" as MCAR or MAR with necessary conditioning variables in the model will not result in bias, merely loss of precision.

Line 755: Similarly, I'd add "non-informative" before "missing data" here, although MI is also useful for scenarios involving MNAR.

Line 865: As a kindness to the reader checking their calculations, you could add "but including the null model" here.

Line 884: I'd personally see all subsets as containing five models here, ~1, A, B, A+B, and A*B.

Line 888: *above*

Line 1011: I'd be inclined to drop point 7 here as while I'd expect this at the end of a workshop, for an article, this is introducing an effectively new topic (there are references to Bayesian methods but nothing that would explain Bayesian inference to someone who hadn't encountered it before).

Line 1046, 1128: Inconsistent capitalisation of title (I think).

Line 1063, 1092, 1099, 1130-1131, 1168-1169, 1224, 1227, 1229: Missing italics.

I'd like to thank you (as well as the reviewers) for your constructive responses. I feel that we are very close to finalising this manuscript!

Reviewer 1 ·

Basic reporting

None

Experimental design

None

Validity of the findings

None

Additional comments

I have read the MS again and the point-by-point replies of all the comments. I support the authors in terms of this paper being published as one. This will be an extremely useful paper for ecologists like myself and my students.

Reviewer 2 ·

Basic reporting

Appropriate.

Experimental design

N/A

Validity of the findings

N/A

Additional comments

This is a paper that I reviewed previously. I thank the authors for their efforts in responding to my comments; all matters raised in my previous review are now addressed to my satisfaction. The current paper presents a nice overview of issues associated with fitting and drawing inference from GLMMs, which readers will find useful.

Reviewer 3 ·

Basic reporting

as before

Experimental design

as before

Validity of the findings

as before

Additional comments

General comments

I had reviewed this manuscript already twice, and I have to say, that the authors largely dealt satisfactorily with my previous comments.

However, I still see some issues regarding the random slopes question. First, random slopes are applicable for covariates and factors (qualitative predictors) and the question which to include and the consequences of neglected random slopes are equally relevant in either case. The para about random slopes (L 415-438), however, leaves the wrong impression as if random slopes are only applicable for covariates. This needs to be fixed. Furthermore, the para still somewhat leaves the impression that the inclusion of random slopes is a matter of taste but after having carefully read the papers Barr et al. 2013 and Aarts et al. (2015. BMC Neurosci, 16, 94), I have to say it is not, but simply a must.

The other issue I still see is use of the term 'model selection' (L 763-981). What the authors leave complete open is the question of what is the purpose of it? I think this is an important question since there is really so much confusion about 'model selection' that I meanwhile came to the conclusion that the title of Burnham's and Anderson's book was badly chosen (the book itself I like lot). In fact, one sees so many papers that utilize model selection to identify a 'best' or 'minimum adequate model' and then test it (and just yesterday some colleagues of mine got a manuscript rejected because they were *not* doing this)... But this is just what statisticians refer to as a 'fishing expedition'. So without some explicit clarifications regarding the purposes of 'model selection' (and the 'no-goes' in this context) I feel the manuscript is incomplete and likely to contribute to misunderstandings prevailing.

Specific comments

L 44: remove 'best practice'?

L 102-103: I'd reserve the acronym GLM for Generalized (i.e., non-Gaussian) Linear Models.

L 194-195: body mass is not 'moved into' but 'added to' the random effects structure (since its fixed effect is kept in the model).

L 196: '… the correlation between the two *and also the fixed effect of body mass*'.

L 201: I'd cite also Barr et al. 2013 and Aarts et al. (2015. BMC Neurosci, 16, 94).

L 293-302: another example would be a model with a miss-specified random slopes structure (i.e., missing random slopes).

L 299-302: according to Barr et al. (2013) a fixed effect is most reliably tested comparing the fit of the full model with that of a reduced model lacking the fixed effect (but being otherwise identical) using a likelihood ratio test. In such a case, the question of residual degrees of freedom is obsolete.

L 319: I'd say such transformations 'aim at' (not 'ensure') the variance being homogeneous.

L 329: 'include*, for instance,* Poisson-distributed counts' (counts can form many other distributions).

L 331: 'or' rather than 'and'.

L 409: 'factors' rather than 'groups'.

L 408-412: the whole concept of 'variance' (in the way its is used in Gaussian models) isn't really applicable in most non-Gaussian models – so I'd replace 'interaction variance' by 'interactive effects' throughout this section.

L 415: random slopes are equally relevant and applicable in case factors (i.e., qualitative predictors). This is an important point which must not be missed! And the entire para must be adjusted accordingly.

L 418: they to not 'allow' but 'force' the fixed effect to 'have a common slope across all experimental units'.

L 421 & 323: Schielzeth & Forstmeier (2009) and Barr et al (2013) (and also Aarts et al. 2015) found clear and convincing evidence for these statements made; as such I find the term 'argue' misleading as it implies these just being opinions (which one could share or not).

L 434: this *likely' inflates Type I error rate.

L 435-438: this is too imprecise. A model can comprise, random intercepts, random slopes, and the correlations among them (but it can also comprise only random intercepts, only random slopes, or only random intercepts and slopes). The model the authors speak about here comprises all the three (i.e., random intercepts, random slopes, and the correlations among them) and only in such a model these correlations can be inspected. And in case of perfect correlations one could still fit a model comprising random intercepts and random slopes but not the correlations among them.

L 439-443: I'd also refer to the papers by Barr et al. and Aarts et al..

L 473: 'predictors' rather than 'data'?

L 596 & 599: I think one usually plots residuals versus fitted values.

L 660: delete 'coefficient'.

L 670: 'compl*e*mentary'.

L 720, 721 & 722: I'd write 'response variable' rather than 'data'.

L 726: delete 'test' ('test statistic' has a very specific meaning; but what is meant here is more generic).

L 747: 'it is *often* not possible'.

L 773-776: when just two models are compared, one lacking a term, this is isn't model selection but simply a way of getting a P-value for the term in question. Also, I would refer to the simpler model as a 'null model' (in the view of many researchers a 'null model' is an intercept only or intercept plus random effects model).

L 932: 'invalid results' in what sense?

L 967: 'uncertainty *in* the'.

L 990: 'unit*s*'.

---

## Author Rebuttal · Round 0.3

**We would like both the editor and the reviewers for continuing to work with us on this manuscript. We particularly appreciate the thoughtful comments pertaining to precision of language and terminology, which we have now worked hard to address in the revised version. With the help of the reviewers we now feel this manuscript has been even further improved as an introductory guide to the world of mixed effects modelling for biologists. We have responded to each individual comment below.**

# Editor's Comments

MAJOR REVISIONS

I would like to thank the reviewers for yet again providing extremely thoughtful and useful comments on this manuscript. I feel that the authors have set themselves a very ambitious task in covering so much material in a single manuscript, even with possible future manuscripts to flesh out aspects that cannot be fully addressed here, in a technically correct manner whilst preserving the necessarily readability for their audience, and responding to the reviewers' comments will help in achieving this goal.

While your preference for a single manuscript is clear, you may of course wish to revisit the option to split this manuscript into two separate manuscripts in your response at this stage. However, I also appreciate that we risk running in circles over the same ground with this, and perhaps some other topics, and so I will ask that you carefully consider all of the reviewers' points and for each, either make changes or provide a justification for not making changes. I agree with the reviewers that the revised version of the manuscript is stronger and hope that it will become stronger still with further revision.

I will draw the authors' attention in particular to the (major) points 2, 3, and 4 from the third reviewer, which need careful consideration, alongside carefully addressing reviewer two's first point. Reviewer two's third point raised an important practical issue (ID labelling) that I think is well worth addressing as I regularly encounter PhD candidates who are unclear whether they are numbering within or across clusters. Needless to say, all of the three reviewers have provided valuable feedback and all of their comments should be addressed.

I think that, in general, you have done well to avoid many overly definite statements in the manuscript, but at line 121, I wonder if a caveat along the lines that "As with all heuristics, there will be situations where these recommendations will not be optimal. If the researcher has concerns about the appropriateness of a particular strategy for a given situation, they should consult with a statistician who has experience in this area." would be useful to remind the reader that relatively few rules, if indeed any, in statistics are universal and to encourage them to collaborate with specialists as appropriate. This would not, of course, relax requirements around technical correctness, but it would, I hope, alert the reader to proceed with due care in this challenging area.

>>>We have added an edited version of this sentence at the end of the first paragraph as requested "As with all heuristics, there may be situations where these recommendations will not be optimal, perhaps because the required analysis or data structure is particularly complex. If the researcher has concerns about the appropriateness of a particular strategy for a given situation, we recommend that they consult with a statistician who has experience in this area."

>>>We appreciate that some of the reviewers strongly advocate splitting manuscript into two and have responded to Reviewer 3 on this issue below. We do not wish to split the manuscript and draw the editor's attention to the overwhelmingly positive feedback we have received from scientists on the two preprint versions of the manuscript. To date the preprint has had nearly 3000 visitors and over 1700 downloads, and we expect this substantial interest in the paper to tield a large number of citations. The manuscript was always meant to be an all-inclusive roadmap to both mixed effects model fitting and methods of model selection, leaving the reader to seek out finer detail elsewhere, and we believe the manuscript functions best in this format.

We have made thorough revisions to the manuscript in line with the extensive and helpful reviewers comments.

# Reviewer 1 (Anonymous)

## Basic reporting

no comments

## Experimental design

no comments

## Validity of the findings

no comments

## Comments for the Author

I have reviewed an earlier version of this MS. The authors have addressed my comments and it seems they did so for others' comments. I have noticed that one of the other referees raised a point about missing data, which I did not comment on. Actually, missing data on mixed models are still underdeveloped and an emerging research topic. There are a couple of papers that directly discuss missing data in mixed model contexts, but which are not currently cited in the MS but relevant.

Nakagawa, S. (2015) Missing data: mechanisms, methods and messages In: Ecological Statistics: contemporary theory and application (eds. Fox, G. A., Negrete-Yankelevich, S. & Sosa, V. J.). Oxford University Press, Oxford. pp. 81-105

Noble, D. W. A. & Nakagawa, S. (submitted) Planned missing data design: stronger inferences increased research efficiency and improved animal welfare in ecology and evolution. bioRxiv

https://www.biorxiv.org/content/early/2018/01/11/247064

The latter one includes an interesting example of how planned missing data design can be used in a repeated measurement design (which is modelled by mixed models). I thought the authors may be interested. No need to cite these but the

authors would like to note that missing data on mixed models are topics under development.

**>>>Thank you for your continued work on the manuscript. We have added both papers to the further reading of the 'Missing Data' section**
* * *
# Reviewer 2 (Anonymous)

## Basic reporting

Language and presentation are appropriate

## Experimental design

Not applicable

## Validity of the findings

Not applicable

## Comments for the Author

In this paper, which I reviewed previously, Harrison et al. provide an overview of considerations associated with (generalised) linear (mixed) modelling in ecological datasets. Overall the paper has much improved from the previous version, and will likely make a valuable contribution to the literature. I feel that a few of my previous comments still apply to the new version, so I hope the details below clarify my concerns for the authors:

1. Regarding the simulation analysis, I accept the authors' response about choosing not to run more models. However, I am still concerned about the mixing of NHST and IT philosophies in the interpretation. Specifically, there is heavy reliance on interpreting results as "Type I error" – strictly speaking one cannot (or perhaps,

should not) falsely reject a true null hypothesis with an information theory approach, because one is not doing null hypothesis significance testing. Of course, I agree with the authors that it is important to consider how often one might draw a false conclusion about whether a given variable is an important predictor of a response. However, it is important not to assess models built under IT using NHST. A better approach would be for the authors to use the inference strategies they advocate elsewhere in the paper, such as testing model assumptions, reporting R2, and basing interpretation on the presence and size of effects in the final model (see also my comment below on RI) – would these metrics collectively lead a researcher to erroneously conclude that they had settled on an appropriately parameterised final model that suggests strong effects? Similarly, the sentence at L944-945 presents a very frequentist viewpoint on model selection, which is inappropriate in a discussion of information theoretic approaches to model selection.

**>>> In response to this comment and that from Reviewer 3, we have now removed the simulations from the manuscript. We have also adjusted our terminology with respect to 'Type I error' to avoid giving the impression that we are mixing analysis paradigms of information theory and frequentist statistics. We now refer to the inclusion of uninformative parameters following the terminology in Arnold 2010 J Wildlife Management.**

2. Regarding the use of delta-6 or 95% summed weights, I felt that the authors response to my previous comment missed my point somewhat, which is that delta values and summed weights represent two (albeit slightly) different approaches. If one wants to "guarantee" 95% weight, then they should use 95% summed weight. For example, in the paper cited by the authors, model "M4" required delta-6.5 to encompass the best EKLD model 95% of the time (Richards, 2008, p223). While "at least delta-6" may be a good general guideline, this is not the same as "guaranteeing 95% weight". See also Burnham and Anderson, 2002, p78, for explanation of how delta values represent relative evidence.

**>>>We have removed the word 'guarantee' from the section and reworded it slightly, as we agree that it gives the impression that delta-6 will always give at least 95% summed probability/weights. We have also highlighted that using 95% summed weights is a slightly different approach to using delta-scores.**

3. New material has been added on the specification of random factors: I believe

the lme4 specification "Clutch Mass ~ Foraging Rate + (1|Woodland) + (1|Female ID)" should achieve the same goal as the model on L402 (should this be M5?), if all female IDs are unique. Ambiguity can arise if the females are labelled (say) 1 – 10 in woodland 1, and 1 – 10 in woodland 2: are there 10 females, each breeding at two sites, or 20 females? Labelling females as 1 – 20 (if the latter) alleviates the ambiguity between crossed and nested random factors, and it would be worthwhile explaining this. See for example lme4.r-forge.r-project.org/book/Ch2.pdf

**>>> We thank the reviewer for highlighting the opportunity to talk about uniquely labelled factor levels for random effects. We have now added a section on this topic:**

**"We advocate that researchers always ensure that their levels of random effect grouping variables are uniquely labelled. For example, females are labelled 1 – *n* in each woodland, the model will try and pool variance for all females with the same code. Giving all females a unique code makes the nested structure of the data implicit, and a model specified as ~ (1| Woodland) + (1|FemaleID) would be identical to the model above. "**

4. Considering the revised section 5 (p34), I think the authors are overly dismissive of "relative importance" statistics. Unfortunately, the vernacular definition of "importance" differs somewhat from the technical definition; "relative importance" (sum of weights) informs which parameters are likely to occur in the best model, not their effect sizes. Indeed, IT-based model selection determines two things: the probability that variables should be present in the model (taking model selection uncertainty into account) and, if variables are included, what their effect sizes are. RI can assist inference around the former. I accept that there is discussion about the effectiveness and interpretation of RI in the literature (e.g. Giam and Olden, 2016; Galipaud et al., 2017), but I am concerned that the tone of section 5 might suggest to readers that they should ignore RI altogether. My view is that inference in an information theoretic framework should take into account (or at least, report!) all measures of quantitative evidence about the models, including RI and parameter estimates. For example, a predictor with RI of 1 and a small effect size may or may not be biologically "important", while a low RI would likely indicate poor support, even if an associated effect is large. These statistics are additional tools for inference.

**>>> We agree with the reviewer and have added a couple of sentences to section 5 to encourage readers to report all quantitative evidence pertaining to information theoretic modelling:**
**"However, summed Akaike weights for variables in top model sets still represent useful quantitative evidence; they should be reported in model summary tables, and ideally interpreted in tandem with model averaged effect sizes for individual parameters. "**

5. A note on the suggested model specifications (p5, p6, p8): it is great these are provided, as knowing how to specify a model in R can be very useful to new users. These models are set within a discussion that considers the differences among models, so it would be helpful to see how the model outputs change depending on the model specifications. This could be easily obtained using a "built in" dataset in R, and presented as supplementary material with a brief commentary on how to interpret each output. For example, the output from M1 and M2 could be narrated in terms of the interpretation of output values associated with "group". The same approach could be applied to M3 and M4. As an aside, a more complete discussion of for the use of random slopes would be provided if the authors explain the circumstances under which one would prefer model M4 over a model specified as "glmer(successful.breed ~ 1 + (body.mass|sample.site)", which is sufficiently similar to model M2 that I think it is worth noting (see also e.g. Gelman and Hill, 2007, p259).

**>>>We agree that this would be a useful resource for researchers. However, we are currently working on a more extensive version of the above suggestions as a separate paper designed to help researchers interpret random effects from all of these types of models fitted to complex ecological data. We thank the reviewer for this helpful suggestion and for their continued work on improving the manuscript.**

6. A note on whether removing models from the set makes interpretation of the Akaike weights "difficult" (L1011): as pointed out by Burnham and Anderson (2002, p75), adding and removing models from a set requires weights be recalculated. From a practical perspective, it is straightforward to calculate the weights of any model set by manually making the list of models for further analysis with MuMIn, or by using 'get.models' (MuMIn) on an existing 'dredge' object – see the MuMIn

documentation for more info. In fact, wi values can even be manually recalculated relatively easily too (using the formula in Burnham and Anderson, 2002, p75). Explicitly specifying the model set is what the authors are indirectly recommending when they suggest that one should consider whether the "all-subsets selection" approach is appropriate, and so mentioning these methods for re-calculating weights would overcome the "difficulty" and provide readers the flexibility of choosing their own model set.

**>>> We agree that it is simple to recalculate model weights from a user-chosen set of models. Because model weight recalculation is done automatically in the *MuMIn* package when subsetting a model list to a certain delta value, we do not think this is something users will struggle with. This statement actually arose as a result of a discussion between one of the authors (XH) and Shane Richards, who advocated that in the case of applying the nesting rule the model weights should be thrown out as they were essentially meaningless. We have no quantitative proof of this, so added this statement merely as a caution to readers about how weights are interpreted. However, this sentence has caused a lot of problems, both here and in another manuscript, so we have removed it from this version.**

Minor comments

L105 "best practice", in light of the title change, best to change this phrase here too?
**>>>We think the reviewer is referring to the reference to the Zuur data exploration paper, which does actually deal with best practice issues. We weren't using it to refer to our own work, so we have kept the phrase here.**
L369 "checking the assumptions of the LMM or GLMM is an essential step" – given the high level of detail and instruction provided elsewhere in the paper, I felt like this statement needs a sentence or two also providing suggestions for some of the methods/packages that could be used.
**>>> We have added a parenthetical statement pointing the reader to the section 'Quantifying GLMM Fit and Performance' which makes a strong case for distinguishing between model fit and model adequacy, and points to some packages that can be used.**

L641 note that it is also possible for data to be under-dispersed (less variance than expected by chance, i.e. more uniform than chance).

**>>>We have added a sentence to discuss underdispersion and the spaMM package that can fit underdispersion models.**

L719 isn't a Gaussian GLMM a LMM?

**>>> We have changed this to 'Gaussian models'**

L769 a dataset cannot be "made up of missing data" – suggest rephrasing

**>>>We couldn't find the phrase the reviewer refers to but have rephrased some of the sentences in the missing data section where we suspect clarity can be improved.**

L839-840 "there is no 'null'" – I see the authors' point, although one occasionally sees the intercept-only model referred to as the 'null model' – suggest rephrasing.

**>>>We have changed to "because NHST requires a simpler (nested) model for comparison"**

L1062 "data dredging" and "fishing" are no longer major components of the paper, and could therefore be removed from the Conclusions?

**>>>We have removed this sentence in response to this comment, and also because it makes reference to 'false positives' which could be considered use of frequentist language in an IT framework.**

The paper contains a lot of abbreviations – some of which are widely used, and others less so. I suggest the authors consider providing a glossary of abbreviations. Furthermore, many abbreviations that are used only once or twice are probably unnecessary, and the paper would be more readable if the terms were spelt out e.g., "GLS" (L633), "IT-AIC" (L775) "ASS" (L897 and 898).

**>>> We have added long-form versions of these abbreviations where appropriate. Reference to 'ASS' has been removed in the updated version. We have not provided a glossary of abbreviations and instead have tried to use long-form wherever possible.**

Literature cited in this review

Burnham, K.P., Anderson, D.R. (2002) Model selection and multimodel inference: a practical information-theoretic approach. Berlin, Springer.

Galipaud, M., Gillingham, M.A.F., Dechaume-Moncharmont, F.-X. (2017) A farewell to the sum of Akaike weights: The benefits of alternative metrics for variable importance estimations in model selection. Methods in Ecology and Evolution, 8, 1668-1678.

Gelman, A., Hill, J. (2007) Data analysis using regression and multileval/hierarchical models. Cambridge, Cambridge University Press.

Giam, X., Olden, J.D. (2016) Quantifying variable importance in a multimodel inference framework. Methods in Ecology and Evolution, 7, 388-397.

Richards, S.A. (2008) Dealing with overdispersed count data in applied ecology. Journal of Applied Ecology, 45, 218-227.

# Reviewer 3 (Anonymous)

## Basic reporting

all okay to good.

## Experimental design

doesn't really apply with the exception of the simulation which is okay, but doesn't really much and conflates mode selection with testing (see my comments to the authors)

## Validity of the findings

doesn't apply

## Comments for the Author

General comments

This manuscript greatly improved as compared to the last version, and I appreciate the authors' efforts to deal with my comments and recommendations. However, I still see need for improvement and refinement. More specifically I see four major points (in my view issue two and four are the most crucial ones):
**>>>Thank you for your careful attention to the manuscript, and especially for identifying several instances where the clarity of our terminology/phrasing could be improved. We are especially keen to avoid confusing readers by looking like we advocate mixing analysis paradigms, and appreciate the reviewer continuing to work with us to ensure our language is precise**

* First, I still feel that the manuscript's two major topics, GLMM and MMI, are not really connected, neither regarding statistical theory nor in the manuscript itself, and that the manuscript would benefit from being split into two. Moreover, I even see dangers in presenting both approaches in the same manuscript. In fact, as the authors themselves (correctly) state, the number of estimated parameters (degrees of freedom) in GLMM are generally unknown (L 883-884). This, in turn poses the question of how to even determine an information criterion such as AIC or BIC which penalizes for model complexity. The authors do not elaborate on this question, beyond briefly mentioning the issue. In their cover letter the authors state that they want to present the entire 'analysis pipeline'; but to me it seems that this particular pipeline (i.e., GLMM in combination with MMI) does not rest on a solid theoretical basis (and the fact that the two methods are frequently combined in applied statistics in ecology and behaviour doesn't alleviate the problem). So maybe better not presenting as being logically following one another, perpetuating common practice but not promoting best practice.
**>>>We agree that combining both topics into a single manuscript is no small task but differ in our opinion with the reviewer on whether the topics are unrelated.**

**Indeed, the Bolker et al 2009 TREE paper discusses both model specification and ways of selecting among those models, including how to calculate the appropriate degrees of freedom for models. We have based our structure on this paper, but updated it for the statistical tools and packages now available**

**nearly 10 years later. We certainly do not feel that it is dangerous to present both topics in a single manuscript (as Bolker et al. also did). In fact, we would argue the converse – that simply dealing with model specification but leaving (novice) readers without an adequate roadmap on how to begin to select among models would itself be dangerous. We do agree the issue(s) of model selection could easily form an entire paper on their own to deal with each issue in depth, but of course doing so was never the original goal of this manuscript. We feel that we have achieved our aim of providing a guide to the entire analysis pipeline and pointing the reader to the resources needed to truly understand each step of the pipeline in more detail.**

**We hope that the reviewer will be understanding of this difference of opinion, and we are very grateful for their continued help and advice on this manuscript**

* Second, I still see need for refinement regarding the sections about random slopes (particularly L 379-402). Although the authors have considerably improved the manuscript regarding this point, the current manuscript still cuts the issue too short. The key point is that the authors seem to mix random slopes and correlations among random intercepts and slopes. In fact, the maximal model proposed by Barr et al. (2013. J Memory Lang, 68, 255–278) represents a model comprising random intercepts, all possible random slopes and also all correlations among them. The authors of the current manuscript are correct in stating that (nearly) perfect correlations are indicative of a model being too complex for the data at hand. However, the situation is not an all-or-nothing. In fact, when the correlations cannot be reasonably estimated, it might still be possible to fit a model including random intercepts and all random slopes but not the correlations among them (see Bates et al. 2015. Journal of Statistical Software, 67, 1-48). This is also the approach somewhat proposed by Barr et al. (2013) and Bates et al. (Parsimonious Mixed Models. http://arxiv.org/abs/1506.04967v1): begin with the most complex model but simplify it, beginning with the exclusion of correlations among random intercepts and slopes when it appears too complex.

There is one additional point I feel the authors must make: it should not be a matter of taste whether random slopes are considered. Instead, based on the available

literature Schielzeth & Forstmeier (2009. Behav Ecol, 20, 416-420), Barr et al. (2013), and Aarts et al. (2015. BMC Neurosci, 16, 94) it simply seems needed to account for random slopes to achieve the nominal type I error rate (and correspondingly unbiased standard errors and confidence intervals); and when all or several random slopes are neglected because the model would get too complex to fit it, then one has to face a perhaps highly elevated type I error risk. Also at other occasions (e.g., L 172-187), the authors still leave the impression that decisions regarding whether to include random slopes are a matter of 'taste' or the aims of the study, but in my view Schielzeth & Forstmeier (2009), Barr et al. (2013), and Aarts et al. (2015) made convincing cases that these have to be included to prevent type I errors.

**>>>We have removed the sentence in the indicated section that suggests that the decision to fit random slopes can depend on the goals of the analysis. We have also added a sentence in the 'Random Slopes' section to acknowledge that removing the correlation between intercepts and slopes is an option. We think this section now gives appropriate advice on how to specify random effects structures, and where to find more detailed information on these topics.**

* Third, I feel concern about the section headed 'Stepwise Selection, Likelihood Ratio Tests and P values' (716-750). In my few this conflates a couple of issues in an inappropriate and confusing way. First, NHST is one of the three major statistical 'philosophies' (together with Bayesian and information theory based inference) used to draw inference about a model and/or individual predictors. Stepwise procedures, in turn, are a specialized technique which had been (and unfortunately still is) used as a means of model simplification; stepwise can be based on NHST, but also on an information criterion like AIC or other criteria (e.g., an F value). As such NHST has nothing to do with stepwise procedures, and the far, far majority of uses of NHST have nothing to do with stepwise selection. As the authors correctly state, both have come under heavy criticism, but for very different reasons. NHST is criticised since decades for the many weaknesses and pitfalls the approach has; stepwise has been criticised for a couple of particular weaknesses (e.g., Whittingham et al. 2006. Journal of Animal Ecology 75, 1182–1189) among which is an extremely inflated type I error probability (Mundry & Nunn. 2009. Am. Nat., 173, 120-123).

**>>> We agree with the reviewer that stepwise inference and NHST are separate entities. We made sure to state this clearly in the original manuscript, by first defining NHST and then defining that stepwise selection can use the NHST framework. We realise that the final paragraph of the section could be interpreted as attributing the flaws of stepwise procedures to the use of NHST, which is misleading for (especially novice) readers, and we thank the reviewer for pointing this out. We have edited this section to remove this issue.**

**Use stepwise AIC but is far less common and so we did not give it any treatment in the previous version, which was an oversight. We have now edited this section to separate NHST from stepwise procedures and acknowledge that some people use stepwise AIC.**
**Of course, discussing stepwise procedures in general is difficult because one must then avoid language like 'significant' predictors, as this phrase doesn't apply to stepwise AIC. So we have added a caveat that we focus on stepwise NHST procedures so that we can be consistent with language.**

Here are a couple of clarifications which seem needed:
- NHST, when not coupled with stepwise and used appropriately in the sense of Forstmeier & Schielzeth (2011. Behav. Ecol. Sociobiol., 65, 47–55) does not lead to overestimated effect sizes or an inflated type I error rate.
- the null model sensu Forstmeier & Schielzeth (2011) is not one that differs from the full model by a single predictor, but by a set of predictors and the comparison between the two reveals a global test of their combined effect (appropriately accounting for multiple testing which otherwise would be an issue in case one would base inference on the significance of the individual terms in the model without conducting a full-null model comparison).
**>>>We agree with the reviewer here, and have mentioned this in the 'Global Model Reporting' section. We do not think we gave the impression in the 'Stepwise selection' section that the F&S 2011 null model was a single-predictor-difference model. We have now moved the 'global model' section up to be included in the 'stepwise' section as they are complementary.**

- Mundry (2011) did not argue in favour or model simplification.

**>>>We have removed the phrase 'model simplification'.**

- Murthaugh (2009) found that stepwise and all subsets approaches revealed largely comparable results (wrt their predictive performance and number variables selected) and regardless of whether stepwise was based on an F-test or information criteria. As such the article does not really compare NHST with other philosophies but compares model selection techniques.

**>>>Thank you for drawing our attention to this. We have removed this sentence.**

"

* My last major concern is about the fact that the authors lack clarity and rigour regarding the issue of mixing model selection with significance testing. In fact, Burnham & Anderson (2002) have repeatedly pointed out that mixing the two philosophies is a no-go (and Mundry 2011 showed that it leads to drastically inflated type I error rates). For instance, when applied appropriately, there is no such thing as a type I error when using model selection based on an information criterion (such as AIC) because in the context of such an analysis one simply must not use any tests (see Burnham & Anderson (2002), e.g., P 202-203). In the view of the authors of the current manuscript, however, such an option seems to exist (see, e.g., L 805-807; 850-852; 855-859; caption of Fig. 4). Its worth emphasizing here that the term 'test', as I used it here, also encompasses checks of whether confidence intervals comprise the zero (and that bias in P-values gets along with parallel biases in standard errors (being too small) and effect sizes (being too large)).

**>>> We thank the reviewer for identifying this inconsistency in our language. We have now changed the phrasing used here to refer to inclusion of uninformative parameters, rather than presence of Type I errors. We state again for emphasis that in no way were we intending to advocate mixing of analysis paradigms.**

Apart from that, I am still not very convinced by the simulation (L 813-836), since it conflates a couple of issues, namely (i) combining model selection with significance testing (see also my previous comment), (ii) conducting pair-wise comparisons without a global test of the effect of a factor, and (iii) neglecting the full-null model model comparison). Each of these have been addressed more thoroughly and

clearly in other papers (i: Burnham & Anderson 2002; Mundry 2011; ii: all stats books covering ANOVA; iii: Forstmeier & Schielzeth 2011), and I feel this section does not contribute much to the existing literature nor to the clarity of the manuscript. On the other hand, the authors should make clearer statements about not combining model selection and significance testing (in a wider sense, i.e., encompassing also inspection of whether confidence intervals encompass zero; see also my previous comment).

**>>> We agree with the reviewer that these simulations conflate several sources of error in a single outcome, and cannot partition the observed rates of uninformative parameter inclusion among them. As such we have decided to remove the simulations from the updated draft of the manuscript to avoid causing unnecessary confusion for the reader.**

Specific comments

L 33: Pretty abrupt transition from one topic to another.
**>>>We have edited the abstract to reflect all the edits made in this version, and so have changed this sentence.**

L 67-68: inference 'about'.
**>>>Changed**

L 69-72: I felt these examples are confusing: if the fixed effect varies or is manipulated at the clutch level, how can pseudo-replication happen on the chick level (it happens on the clutch level if each comprises several chicks, each measured once). And 'if fixed effects vary at the level of the chick': between or within chick? In case of the former the authors are correct that non-independence occurs among clutches or mothers'; but in case of the latter it also happens on the chick level.
**>>>We have clarified the wording of this section:**
**"In our example, if the fixed effect varies or is manipulated at the level of the clutch, then treating multiple chicks from a single clutch as independent would represent pseudoreplication, which can be controlled carefully by using random effects."**

L 145: 'hierarchically' implies that certain effects get priority when being estimated; but to my knowledge all effects are modelled/estimated simultaneously.

**>>>We agree that all effects are estimated simultaneously. Our point was to emphasise the hierarchical structure of variance components in such models. We do not think that this section implies temporal ordering of component estimation, and in fact removed any misleading references to this in the first revision of the manuscript. We have not made any changes here.**

L 165-167: in my view random slopes need to be included to keep type I error rate at the nominal level of 0.05 and obtain unbiased standard errors and confidence intervals.

**>>>We have removed this sentence now to prevent readers thinking it is a choice. We discuss the random slopes paper a few lines below this and have updated the other section in the manuscript in line with the comments above.**

L 179-182: it seems worth mentioning that this model comprises the random intercept, the random slope and also the correlation between the two.

**>>>We have now added this as requested**

L 183: Schielzeth & Forstmeier (2009) *show* that...

**>>>Changed**

L 274-279: I don't get the point here. Certainly, the random effects structure needs to be set up appropriately; but as Barr et al. (2013. J Memory Lang, 68, 255–278) showed, P-values for individual effects should best be based on likelihood ratio tests comparing the full with respective reduced models. This, in turn, doesn't rely on determination of residual degrees of freedom.

**>>>If one were performing F tests for models fitted to a Gaussian trait, then determination of the residual degrees of freedom is central to the test to derive a p value. We have noted this particular misuse of statistical tests between full and reduced models when such pseudoreplication is present.**

L 285-287: *general* linear models make the assumptions of normality and homogeneity of residuals (in my view, 'linear models' is a generic term

encompassing also Generalized Linear Models such a Poisson or logistic models).
**>>>We respect that this may be the view of the reviewer, but we don't believe it is universally shared. However, we have changed the syntax as requested.**

L 287: lower case 'normality'.
**>>>Changed**

L 308-318: my intuition tells me that those not being already familiar with the link function wouldn't get this section.
**>>>We agree that link functions are a tricky topic for scientists to get to grips with initially. We deliberately point the reader to appropriate texts**

L 372-376: but this applies in general (not only to GLMM).
**>>>We have changed this to '(G)LMMs'**

L 374: 'Crossed factors *allow to* accurately estimate...'.
**>>>We have changed to "Crossed factors allow the model to accurately estimate…"**

L 566: Zuur et al *give*.
**>>>Changed. Thank you for pointing this out.**

L 586: 'estimates *of* …'.
**>>>Changed**

L 658-659: to my knowledge, Wald-/t-tests are not applicable for random effects (and maybe not even F-tests).
**>>>We have changed this section to simply say 'for tests of random effects'.**

L 694: 'often' or 'always'.
**>>>We have changed this sentence to "When collecting ecological data it is not possible to measure all of the predictors…"**

L 779: 'reference*s*'.
**>>>Changed**

L 788: '...details on *how* AIC...'.
>>>**Changed**

L 792-795: comparing the full with a null model is not an alternative to NHST, but simply NHST applied appropriately (avoiding type I errors due to multiple testing)..
>>>**We have changed this sentence to "Performing 'full model tests' (comparing the global model to an intercept only model) before investigating single-predictor effects controls the Type I error rate…"**

L 805-807: but this applies only when one selects the best model and then tests it, something which should never be done as repeatedly and strongly stated by Burnham and Anderson (2002).
>>>**We have changed the wording here to say that this approach increases the risk of including uninformative parameters, following the language used by Arnold (2010). We certainly weren't advocating testing the top model, but can see that the language we used may have made it seem that way.**

L 817: 'ASS' not formally introduced.
>>>**We have now removed this section pertaining to the simulations that mentioned ASS**

L 818: I guess you the authors mean 'different' rather than 'separate'.
>>>**This section has now been removed.**

L 817-819: I don't see any difference: what the function dredge of the package MuMIn does is exactly dredging in the sense of fitting all possible subsets of models.
>>>**This section has now been removed.**.

L 849-852: as Burnham and Anderson (2002) have pointed out repeatedly one should not use the term 'significant' in the context of an AIC-based analysis (e.g., P 84, 203), and I feel the authors should adhere to this rule.
>>>**We apologise again for imprecise language. The authors all believe firmly**

**in not mixing analysis paradigms and do not wish to give the impression that we do. We have changed this phrase to 'uninformative parameters'**

L 855-860: another example of the authors mixing model selection and significance testing. According to Burnham and Anderson (2002) such exercise must not be done (see previous comment), and Mundry (2011) clearly showed how such practice leads to drastically inflated type I error rate.
**>>>As above, we apologise for imprecise language and have removed this sentence, especially as it pertains to the removed simulations**

L 859: which 'conditions'?
**>>>We have removed this sentence.**

L 930: 'included' in what? The AIC-best model?
**>>>We have changed this to "included in the top model set"**

L 936: I'd use 'similar' rather than 'equal'.
**>>>Changed**

Figure 1, caption: reference to (B) is missing. The fact that the overall intercept is 0 is irrelevant here.
**>>>We have now added the missing figure caption. Thank you for pointing this out. We have also removed the reference to the intercept being zero.**

Figure 2, caption: I'm not sure if I would speak of a biased estimate when its average seems pretty much perfectly matching the simulated value (I'd speak of 'bias' then the average deviates from the simulated value).
**>>>We have changed the phrasing to "With moderate collinearity, estimation of $\beta_{x1}$ is precise, but certainty of the sign of $\beta_{x2}$ is low. When collinearity is strong, estimation of $\beta_{x1}$ is far less precise, with 14% of simulations estimating a negative coefficient for the effect of *x1*"**

---

## Round 0.4 · accepted · Accept

As has been the case since its initial submission, this is an ambitious manuscript and I think that with your constructive approach to the reviewers' equally constructive and extremely thorough comments, we have reached the point where the manuscript will provide a valuable reference to both experienced and not-yet-experienced modellers looking at these topics. While there are always more things that we would want such researchers to know, and details for them to be better aware of, perhaps there will one day be a sequel manuscript that can cover at least some of this material!

I would again like to thank the reviewers for their exceptional efforts in suggesting areas for improvements to the manuscript. I hope that I can call on all of you in the future for any similar manuscripts.

#